# Analysis of NIS Plasma Membrane Interactors Discloses Key Regulation by a SRC/RAC1/PAK1/PIP5K/EZRIN Pathway with Potential Implications for Radioiodine Re-Sensitization Therapy in Thyroid Cancer

**DOI:** 10.3390/cancers13215460

**Published:** 2021-10-30

**Authors:** Márcia Faria, Rita Domingues, Maria João Bugalho, Ana Luísa Silva, Paulo Matos

**Affiliations:** 1Serviço de Endocrinologia, Diabetes e Metabolismo do CHULN-Hospital Santa Maria, 1649-028 Lisboa, Portugal; mcfaria@fc.ul.pt (M.F.); rsofiadomingues@fm.ul.pt (R.D.); maria.bugalho@chln.min-saude.pt (M.J.B.); analuisasilva@fm.ul.pt (A.L.S.); 2BioISI-Biosystems and Integrative Sciences Institute, Faculdade de Ciências da Universidade de Lisboa, 1749-016 Lisboa, Portugal; 3Departamento de Genética Humana, Instituto Nacional de Saúde Doutor Ricardo Jorge, 1649-016 Lisboa, Portugal; 4ISAMB-Instituto de Saúde Ambiental, Faculdade de Medicina da Universidade de Lisboa, 1649-028 Lisboa, Portugal; 5Serviço de Endocrinologia, Diabetes e Metabolismo, CHULN and Faculdade de Medicina da Universidade de Lisboa, 1649-028 Lisboa, Portugal

**Keywords:** thyroid cancer, RAI-refractory, NIS, RAC1 signaling, plasma membrane localization

## Abstract

**Simple Summary:**

The presence of the sodium–iodide symporter (NIS) at the plasma membrane (PM) of differentiated thyroid cancer (DTC) cells is required for the successful use of radioiodine (RAI) therapy in these malignancies. However, NIS is frequently downregulated in malignant thyroid tissue, and 30% to 50% of metastatic DTCs become refractory to RAI treatment. Notably, many RAI-refractory DTCs still show relatively abundant NIS protein levels despite failing to uptake sufficient iodide to enable RAI therapy. This suggests an impairment in the NIS delivery and retention at the PM. Because little is known about these mechanisms, we employed a new experimental strategy to purify and characterize proteins which are selectively associated with NIS at the PM. We found that, when at the PM, NIS is particularly associated with proteins that regulate the cell’s actin cytoskeleton. Further investigation revealed that several of these proteins integrate an intracellular signaling pathway that modulates the functional localization of NIS at the PM, in normal and DTC cells. Our findings open new venues for therapeutic intervention in refractory DTC, envisioning the re-sensitization of these tumors to RAI therapy.

**Abstract:**

The functional expression of the sodium–iodide symporter (NIS) at the membrane of differentiated thyroid cancer (DTC) cells is the cornerstone for the use of radioiodine (RAI) therapy in these malignancies. However, NIS gene expression is frequently downregulated in malignant thyroid tissue, and 30% to 50% of metastatic DTCs become refractory to RAI treatment, which dramatically decreases patient survival. Several strategies have been attempted to increase the NIS mRNA levels in refractory DTC cells, so as to re-sensitize refractory tumors to RAI. However, there are many RAI-refractory DTCs in which the NIS mRNA and protein levels are relatively abundant but only reduced levels of iodide uptake are detected, suggesting a posttranslational failure in the delivery of NIS to the plasma membrane (PM), or an impaired residency at the PM. Because little is known about the molecules and pathways regulating NIS delivery to, and residency at, the PM of thyroid cells, we here employed an intact-cell labeling/immunoprecipitation methodology to selectively purify NIS-containing macromolecular complexes from the PM. Using mass spectrometry, we characterized and compared the composition of NIS PM complexes to that of NIS complexes isolated from whole cell (WC) lysates. Applying gene ontology analysis to the obtained MS data, we found that while both the PM-NIS and WC-NIS datasets had in common a considerable number of proteins involved in vesicle transport and protein trafficking, the NIS PM complexes were particularly enriched in proteins associated with the regulation of the actin cytoskeleton. Through a systematic validation of the detected interactions by co-immunoprecipitation and Western blot, followed by the biochemical and functional characterization of the contribution of each interactor to NIS PM residency and iodide uptake, we were able to identify a pathway by which the PM localization and function of NIS depends on its binding to SRC kinase, which leads to the recruitment and activation of the small GTPase RAC1. RAC1 signals through PAK1 and PIP5K to promote ARP2/3-mediated actin polymerization, and the recruitment and binding of the actin anchoring protein EZRIN to NIS, promoting its residency and function at the PM of normal and TC cells. Besides providing novel insights into the regulation of NIS localization and function at the PM of TC cells, our results open new venues for therapeutic intervention in TC, namely the possibility of modulating abnormal SRC signaling in refractory TC from a proliferative/invasive effect to the re-sensitization of these tumors to RAI therapy by inducing NIS retention at the PM.

## 1. Introduction

Iodide (I^−^) accumulation in thyroid follicular cells constitutes the first step for the biosynthesis of the iodine-containing thyroid hormones [1]. The I^−^ uptake into thyroid cells is mediated through the active transport by the Na^+^/I^−^ symporter (NIS), an integral plasma membrane (PM) glycoprotein located at the basolateral membrane of follicular cells [2,3]. Active NIS-mediated I^−^ transport is electrogenic, and relies on the driving force of the Na^+^ gradient generated by the Na^+^/K^+^ ATPase to simultaneously transport one I^−^ and two Na^+^ ions into thyroid cells [2].

The NIS expression at the PM of thyroid follicular cells is not only important for thyroid hormone biosynthesis but also constitutes the cornerstone for radioiodine (RAI) therapy for differentiated thyroid cancer (DTC) [4,5,6]. Indeed, the selective use of RAI, based on patients’ individual risk, is often recommended as an adjuvant treatment for DTC, for which the standard care includes total thyroidectomy and TSH-suppressive therapy. Moreover, RAI therapy is the first-line treatment for DTC distant metastasis, and is often seen as the most successful targeted internal radiation therapy to date [7].

RAI therapy’s effectiveness is ultimately dependent on the functional NIS expression at the PM of DTC. However, NIS gene expression is frequently downregulated in thyroid cancer [8]. Despite this reduction, most DTCs accumulate I^−^ to some extent, which often allows sufficient RAI accumulation to enable treatment [9]. However, 30% to 50% of metastatic DTCs become refractory to RAI treatment cells, as deficient I^−^ accumulation is the major cause of treatment failure [10,11,12,13], and these patients have considerably lower 5-year disease-specific survival rates [14]. Moreover, those with refractory metastatic thyroid cancer have the worst outcomes, with a 10-year survival rate of roughly 10% [15]. Hence, re-sensitizing refractory tumors to RAI could potentially improve survival for patients with DTC.

Several lines of research are underway to develop new strategies to increase the NIS expression levels in refractory thyroid cancer cells, but most studies have focused on inducing an upregulation of NIS at the transcriptional level [16,17]. However, there are many RAI-refractory DTCs in which the NIS mRNA levels are relatively abundant but only reduced levels of iodide uptake are detected, suggesting an impairment in NIS posttranslational processing, trafficking and/or retention at the cell surface [18,19].

These observations indicate that successful I^−^ re-sensitization approaches have to encompass strategies to enhance NIS transcript levels, its posttranslational regulation and its functional PM residency. Nevertheless, little is known about the identity of the molecules regulating the latter processes, as only three NIS interacting partners have been experimentally validated so far: PTTG1- binding factor (PBF) [20,21,22], ADP-ribosylation factor 4 (ARF4), and valosin-containing protein (VCP) [23]. Ectopic PBF overexpression in thyroid cells posttranslationally represses I^−^ uptake by binding to NIS and leading to its internalization into clathrin-coated late endosomes [22]. ARF4 appears to promote NIS vesicular trafficking from the Golgi to the PM, whereas VCP, a component of endoplasmic reticulum-associated degradation machinery, prevents NIS maturation, possibly by promoting its unfolding prior to proteasomal degradation [23]. Far less clear are the molecules and mechanisms regulating the NIS residency and function at the PM. Several immunohistochemical studies of refractory tumor cells have shown readily detectable levels of NIS protein, but located in intracellular compartments [7]. These observations suggest either a failure in the delivery of NIS to the PM or an abnormal retention and stability at the cell membrane. As such, we ask “What are the molecules and pathways regulating NIS residency at the PM of thyroid cells?”

In order to investigate this issue, we used mass spectrometry to characterize macromolecular NIS complexes, not only from whole cell lysates but also by a specifically developed immunoprecipitation approach to selectively purify the NIS-containing complexes at the PM. This strategy allowed us to further characterize the NIS interactome in thyroid cells, particularly at the PM, and to identify a new SRC-triggered RAC1 GTPase-controlled pathway regulating the NIS functional residency at the cell surface.

## 2. Materials and Methods

### 2.1. Generation of the HA-NIS Construct

A functional full-length NIS construct containing a triple HA tag in the fourth extracellular loop (between nucleotide 969 and nucleotide 1017) was generated by amplifying SLC5A5 (NCBI Reference Sequence NM_000453.2) from pcDNA3.1-C-(k)DYK vector (OHu19270, GenScript, Piscataway, NJ, USA) using the primers NheI-Fw (5′ CTAGCTAGCATGGAGGCCGTGGAGACCG) and NotI-Rv (3′GCGGCCGCTCAGAG GTTTGTCTCCTGCTGG). The amplicon was cloned into the pCR2.1 TOPO-TA vector (Invitrogen, Carlsbad, CA, USA) and subcloned into the pECFP-N1 vector (BD Biosciences Clontech, San Carlos, CA, USA), as a NheI/NotI fragment, by removing the CFP sequence. A fragment of approximately 550 bp, composed of part of the NIS coding region with a hemagglutinin tag (YPYDVPDYA) inserted in triplicate (3HA) in the fourth extracellular loop (between nucleotide 969 and nucleotide 1017), flanked by the PfmlI/BglII restriction sites, was synthesized by a specialized company (NZYTech, Lisboa, Portugal). The final construct, henceforward referred to as HA-NIS, was generated by the exchange of the corresponding part of the NIS coding region from the initial NIS plasmid with the 3HA-NIS synthetized fragment, using the flanking PfmlI/BglII restriction sites. The entire plasmid sequence was confirmed by Sanger sequencing.

### 2.2. Cell Line Generation

The experiments were performed with the commercially available TPC1 cell line (CVCL_6298), originally derived from primary human PTC, and modified to stably express the HA-NIS construct. In brief, the cells were transfected with Lipofectamine 2000 (Invitrogen, Carlsbad, CA, USA), using 2 µg HA-NIS linearized DNA. After 24 h, the cells were trypsinized and transferred at a high dilution to 100 mm dishes containing 1000 μg/mL Geneticin (Gibco, Grand Island, NY, USA). The cells were grown under selection conditions for 15 days, and the resistant clones were scraped, collected and expanded separately on 24-well plates. The clones were screened for NIS expression by Western blot (see below) using an anti-HA (11 583 816 001, Roche, Mannheim, Germany) antibody, and were functionally validated as described in the text. In parallel, CFP-expressing cell clones were generated using similar transfection and selection conditions, in order to be used as control cell lines. The established HA-NIS-TPC1 cell line was further modified to stably co-express the halide sensor YFP-F46L/H148Q/I152L (HS-YFP, a gift from Haggie, University of California, San Francisco [UCSF], School of Medicine). The cells expressing the HS-YFP were selected with 250 μg/mL Hygromycin B (Santa Cruz Biotechnology [SCBT], Santa Cruz, CA, USA). PCCL3 cells expressing HS-YFP were previously described [24].

### 2.3. Cell Culture, Treatment and Transfection

Modified TPC1 cells were cultured in RPMI (Gibco, Grand Island, NY, USA) supplemented with GlutaMAX, 10% Fetal Bovine Serum (FBS, Gibco, Grand Island, NY, USA), and the above-indicated selection antibiotics. The commercial PCCL3 (BCRJ 0204) cell line, derived from Fischer rats’ thyroid follicular normal epithelium, was maintained in Coon’s F-12 modified liquid medium (Merck, Darmstadt, Germany) supplemented with 5% FBS, 10 μg/mL insulin (Sigma-Aldrich, St. Louis, MO, USA), 5 μg/mL Apo-Transferrin (Apo-T) (Sigma-Aldrich, St. Louis, MO, USA) and 0.1 mU/mL TSH (Sigma-Aldrich, St. Louis, MO, USA). When appropriate, the cells were cultured in starvation medium (F12 Coon’s Modification medium supplemented with 0.2% (*v*/*v*) FBS and 5 μg/mL of Apo-T) or stimulation medium (F12 Coon’s Modification medium supplemented with 5% (*v*/*v*) FBS, 5 μg/mL of Apo-T and 1 mU/mL of TSH). All of the cell lines were maintained at 37 °C in a 5% humidified CO_2_ environment, regularly checked for mycoplasm infection, and discarded after 20 passages.

For the plasmid transfections, TPC1 cells at 60–70% confluence were transfected with Lipofectamine 2000, according to manufacturer’s instructions, using a DNA/Lipofectamine ratio of 1:4 (μg/μL) and analyzed 24 h later. Several previously published plasmids were used: pEGFP-RAC1-L61, pCDNA3-MYC-RAC1-V12 [25,26], pECFP-EZRIN [27], pEGFP-CDC42-L61, pEGFP-CDC42-N17, pEGFP-RHOA-L63 and pEGFP-RHOA-N19 [28].

For the RNA interference experiments, the cells were transfected using Lipofectamine 2000 in 35 mm dishes or 8-well chamber slides (Ibidi, Gräfelfing, Germany) with 100 pmol or 10 pmol, respectively, of specific pre-designed siRNAs. The cells were analyzed 48 h after the siRNA transfection, and the silencing efficiencies were verified in each experiment by Western blotting. The siRNA oligos against IQGAP1 (sc-35700), COPA (sc-43696), RAC1 (sc-36351), EZRIN (sc-35349), ARP3 (sc-29739), PAK1 (sc-29700) and PIP5K (sc-36232) were purchased from SCTB (Santa Cruz, CA, USA). A siRNA targeting the exogenous firefly luciferase gene (LUC; 5′-CGU ACG CGGAAU ACU UCG ATT) (Eurofins Genomics, Ebersberg, Germany) was used as a control siRNA.

The treatment of the cells with the RAC1 selective inhibitor EHT1864 (50 μM, SCTB, Santa Cruz, CA, USA), ARP2/3 Complex Inhibitor CK666 (100 µM, Sigma-Aldrich, St. Louis, MO, USA), JNK inhibitor SP600125 (30 µM, Sigma-Aldrich, St. Louis, MO, USA), P38 inhibitor SB203580 (10 µM, SCTB, Santa Cruz, CA, USA), PAK inhibitor IPA3 (10 µM, Calbiochem), SRC Inhibitors PP2 (2 µM, Sigma-Aldrich, St. Louis, MO, USA) or Dasatinib (150 nM, Sigma-Aldrich, St. Louis, MO, USA) was performed for 1 h in the appropriate medium, using cells treated with the same volume of solvent (vehicle) as a control.

### 2.4. Co-Immunoprecipitation of Plasma Membrane-Specific NIS Interactors

HA-NIS-TPC1 cells at near confluence in 100 mm culture dishes (two dishes per condition) were placed on ice in a cold room and washed three times with ice-cold PBS^++^ (PBS pH 8.0 containing 0.1 mM CaCl_2_ and 1 mM MgCl_2_), and were left for 5 min with PBS^++^ to ensure the arrest of endocytic traffic. The cells were incubated with agitation at 4 °C for 2 h, with 4 µg/mL of either rabbit anti-HA (H6908; Sigma-Aldrich, St. Louis, MO, USA) or rabbit anti-goat IgG (5160–2104; Bio-Rad; Hercules, CA, USA; control condition), and were then washed for 3 × 5 min with ice-cold PBS^++^. The cells were lysed in 500 µL lysis buffer [50 mM Tris/HCl pH 7.5, 2 mM MgCl_2_, 150 mM NaCl, 10% (*v*/*v*) glycerol, 1% (*v*/*v*) NP40, protease inhibitor cocktail (Sigma-Aldrich, St. Louis, MO, USA)] and cleared at 10,000× *g* for 10 min at 4 °C. Each lysate was added to 40 µL streptavidin agarose beads, and was rotated for 1 h at 4 °C, as a pre-clearing step to reduce non-specific binding. The pre-cleared lysates were incubated under rotation with 30 µL Dynabeads G-protein beads (Invitrogen, Carlsbad, CA, USA) for 1 h at 4 °C. The beads were then collected by brief centrifugation, and were washed 5 times for 5 min with ice-cold wash buffer (50 mM Tris/HCl pH 7.5, 2 mM MgCl_2_, 300 mM NaCl, 10% (*v*/*v*) glycerol, 1% (*v*/*v*) NP40). The captured proteins were eluted in 50 µL 2× modified Laemmli buffer (62.5 mM Tris/HCl pH 6.8, 3% (*v*/*v*) SDS, 10% (*v*/*v*) glycerol, 0.02% (*v*/*v*) bromophenol blue, 296.4 mM dithiothreitol (DTT)) and stored at −80 °C until they were further analyzed by Western blotting or mass spectrometry.

### 2.5. Co-Immunoprecipitation of the Whole Cell NIS Interactors

HA-NIS-TPC1 cells were seeded in 100 mm dishes and allowed to grow to near 100% confluence. The cells were placed on ice, washed three times with ice-cold PBS^++^, and lysed in 500 µL lysis buffer [50 mM Tris/HCl pH 7.5, 2 mM MgCl_2_, 150 mM NaCl, 10% (*v*/*v*) glycerol, 1% (*v*/*v*) NP40, protease inhibitor cocktail (Sigma-Aldrich, St. Louis, MO, USA)]. The cell lysates were cleared at 10.000× *g* for 10 min at 4 °C. In order to reduce the non-specific binding, the lysates were pre-cleared by incubation with 40 μL G-protein agarose beads (Roche, Mannheim, Germany) for 60 min at 4 °C. The pre-cleared lysates were incubated overnight at 4 °C with 4 µg/mL of either rabbit anti-HA (Sigma-Aldrich, St. Louis, MO, USA) or goat anti-rabbit IgG (Bio-Rad; Hercules, CA, USA; control condition). Next, the lysates were rotated for 1 h at 4 °C with 30 µL Dynabeads G-Protein (Invitrogen, Carlsbad, CA, USA), and the immunoprecipitates were washed 5 times for 5 min with ice-cold wash buffer (50 mM Tris/HCl pH 7.5, 2 mM MgCl_2_, 300 mM NaCl, 10% (*v*/*v*) glycerol, 1% (*v*/*v*) NP40). The proteins were recovered in 50 µL 2× modified Laemmli buffer and further analyzed by Western blotting or mass spectrometry.

### 2.6. Western Blot Analysis

The protein lysates were resolved in 10% SDS-PAGE, except when analyzing the NIS and RAC1 levels, for which 9% SDS-PAGE with 1% glycerol and 12% SDS-PAGE was used, respectively. The proteins were transferred onto polyvinyldene difluoride (PVDF) membranes (Bio-Rad; Hercules, CA, USA) and probed with the appropriate primary antibodies: mouse anti-HA (11 583 816 001; Roche, Mannheim, Germany), rabbit anti-HA (H6908; Sigma-Aldrich, St. Louis, MO, USA), mouse anti-PCNA (NA03; Merck, Darmstadt, Germany), mouse anti-α-Tubulin (T5168; Sigma-Aldrich, St. Louis, MO, USA), rabbit anti-GLUT1 (ab652; Abcam), mouse anti-COPA (sc-398099; SCTB, Santa Cruz, CA, USA), mouse anti-IQGAP1 (sc-374307; SCTB, Santa Cruz, CA, USA), mouse anti-RAC1 (05-389; Merck, Darmstadt, Germany), mouse anti-EZRIN (610602; BD Biosciences;), rabbit anti-GFP (ab290; Abcam), mouse anti-ARP3 (sc-48344; SCTB, Santa Cruz, CA, USA), rabbit anti-NIS (24324-1-AP; Protein-Tech, Rosemont, IL, USA), mouse anti-phospho-JNK (612541; BD Biosciences), mouse anti-phopho-p38 (612565; BD Biosciences), rabbit anti-PAK1/2/3 (2604; Cell Signaling, Danvers, MA, USA), mouse anti-NHERF1 (sc-271552; SCTB, Santa Cruz, CA, USA) and mouse anti-PIPKIα (sc-377021; SCTB, Santa Cruz, CA, USA). The proteins were detected using a horseradish peroxidase-conjugated secondary antibody (Bio-Rad; Hercules, CA, USA) and the enhanced chemiluminescence (ECL) method. When appropriate, the membranes were stripped of the bound antibodies using 250 mM NaOH solution, and were re-probed with additional antibodies. For the densitometry analysis of the WB bands, X-ray films were digitalized, and the images were analyzed with ImageJ software (NIH, Bethesda, MD, USA). Original blots can be found at File S1.

### 2.7. Mass Spectrometry (MS) and MS Data Analysis

Immunoprecipitated NIS-associated proteins samples were sent to peptide mapping by a nanoLC-MS/MS using Sciex TripleTOF 6600 (SCIEX, Framinghan, MA, USA) mass spectrometer, outsourced to UniMS (ITQB, Oeiras, Portugal). The mass spectra were processed using ProteinPilot™ Software v. 5.0 (SCIEX, Framinghan, MA, USA) for the protein identification, and the protein search was performed against the SWISS-PROT protein sequence database (protein sequences from the organism *Homo sapiens*). In each replicate, every identified protein was characterized by a protein confidence score (PCS). This score was calculated by the analysis software, and it reflects the amount of total, unique peptide evidence related to each identified protein. The value of confidence (Cf) in the identification of a particular protein in a sample is given by the equation: Cf (%) = 100 × *(1−10^-PCS). Thus, for PCSs above 1.3, the confidence in the protein identity will be higher than 95%. All of the experiments were performed in triplicate. The selection of high-confidence NIS-interactor candidates among the identified proteins, followed the analytic pipeline, is described in the text. STRING database software (https://string-db.org/, accessed on 3 May 2021) was used to perform the Gene Ontology (GO) and Kyoto Encyclopedia of Genes and Genomes (KEGG) pathway enrichment analysis on the selected high-confidence candidate datasets, as well as to construct the protein networks and generate protein interaction maps.

### 2.8. Biotinylation of the Cell-Surface Proteins

The cells were seeded in 35-mm culture dishes at appropriate seeding densities to allow them to reach confluency at the day of lysis. After each specific treatment, the cells were placed on ice in a cold room, washed three times with ice cold PBS^++^, and left on ice for 5 min to ensure the arrest of endocytic traffic. The cells were then incubated for 30 min with 0.5 mg/mL sulfosuccinimidyl 3-[[2-(Biotinamido)ethyl] dithio] propionate sodium salt (sc-212981, Santa Cruz Biotechnology), a water-soluble, membrane-impermeable biotinylation reagent that allows the selective labeling of all of the proteins at the cell surface. The cells were rinsed twice and left for 15 min on ice with ice-cold quenching buffer [100 mM Tris/HCl pH 7.5, 150 mM NaCl, 1 mM CaCl_2_, 1 mM MgCl_2_, 10 mM glycine, 1% (*w*/*v*) BSA]. The cells were again washed three times with ice-cold PBS^++^, and lysed in lysis buffer [50 mM Tris/HCl pH 7.5, 100 mM NaCl, 10% (*v*/*v*) glycerol, 1% (*v*/*v*) NP40, 0,1% (*v*/*v*) SDS], supplemented with a protease inhibitor cocktail [1 mM PMSF, 1 mM 1,10-phenanthroline, 1 mM EGTA, 10 μM E64, and 10 μg/mL of each aprotinin, leupeptin, and pepstatin A (all from Sigma-Aldrich, St. Louis, MO, USA)]. The cell lysates were cleared at 9000× *g* for 5 min at 4 °C. An aliquot of 40 μL—representing the protein input levels—was removed, while 200 μL lysate was added to 20 μL streptavidin-agarose beads (Sigma-Aldrich, St. Louis, MO, USA), which were previously incubated for 1 h in pull-down buffer containing 2% (*w*/*v*) non-fat milk, and washed three times in pull-down buffer. After 1 h rotation at 4 °C, the streptavidin-beads were centrifuged for 1 min at 3000× *g* and washed three times in cold wash buffer (100 mM Tris/HCl pH 7.5, 300 mM NaCl, 1% (*v*/*v*) Triton X-100). The captured proteins were recovered in 20 μL of 2× modified Laemmli Buffer and analyzed by Western blotting.

### 2.9. Immunofluorescence and Confocal Microscopy

In order to detect the NIS-HA levels at the PM versus the total NIS-HA, a specific immunofluorescence protocol for intact cells was used. In brief, NIS-HA-expressing cells grown on 10 × 10 mm coverslips were rinsed on ice with ice cold PBS^++^ and labeled with rabbit anti-HA (H6908; Sigma-Aldrich, St. Louis, MO, USA) for 1 h at 4 °C, without permeabilization. The coverslips were then thoroughly washed 3 times with an excess of ice-cold PBS^++^, and were incubated with anti-rabbit Alexa Fluor 633 for 30 min at 4 °C. The cells were then fixed with 4% (*v*/*v*) formaldehyde at pH 7.4, for 15 min at 4 °C, followed by 15 min at room temperature, permeabilized with 0.5% (*v*/*v*) Triton X-100 in PBS for 15 min, and incubated with mouse anti-HA (11 583 816 001Roche, Mannheim, Germany) for 1 h at room temperature. The coverslips were then washed with 0.01% (*v*/*v*) Triton X-100 in PBS (PBST) and incubated with anti-mouse Alexa Fluor 488 for 30 min at room temperature (RT). Next, the coverslips were washed 3 times in PBST, their nuclei were stained with DAPI (4′,6-diamidino-2-phenylindole; Sigma-Aldrich, St. Louis, MO, USA), and then the labeling was fixed with 4% (*v*/*v*) formaldehyde for 15 min, followed by mounting on microscope slides using Vectashield (Vector Labs, Burlingame, CA, USA), and sealing with nail polish. The labeled cells were then imaged on a Leica (Wetzlar, Germany) TCS-SPE confocal microscope.

### 2.10. HS-YFP-Based NIS Functional Assay

In order to study the HA-NIS protein function, TPC1 cells stably co-expressing the HA-NIS and HS-YFP proteins were used. For the influx assays, the cells were plated on poly-L-Lysin coated 8-well chamber slides (Ibidi, Gräfelfing, Germany) and allowed to grow to near 70–80% confluence. After treatment or transfection, as indicated, the cells were washed and incubated for 15 min at 37 °C with PBS-influx-solution (137 mM NaCl, 2.7 mM KCl, 0.7 mM CaCl_2_, 1.1 mM MgCl_2_, 1.5 mM KH_2_PO_4_, 8.1 mM Na_2_HPO_4_, and 10 mM glucose, pH 7.4) and transferred to an inverted fluorescence microscope (Leica, Wetzlar, Germany) for the time-lapse analysis. A PBS-NaI influx solution [containing 50 mM NaI (TPC1) or 1 mM NaI (PCCL3)] was added to the cells, and the decay of YFP fluorescence was followed by live imaging for 500 s, acquiring an image every 10 s. In order to confirm that the iodine influx observed was specifically mediated by NIS, replicate assays were performed in the presence of equimolar ClO_4_^−^ (for 10 min, before the PBS-NaI solution was added), a competitive inhibitor of iodide uptake by NIS. The image series were stacked and analyzed with Image J (NIH, Bethesda, MD, USA), defining whole field regions of interest (ROIs) that excluded saturated cells for the pixel intensity measurements. After background subtraction, the YFP fluorescence recordings (F) were normalized to the initial value measured before addition of I^−^ (F_0_). The average fluorescence decay was fitted to an exponential decay function to derive the maximal slope that is proportional to the initial influx of I- into the cells. The maximal slopes were converted into I^−^ variation rates using the equation d[I-]/dt = Kd [d(F/F_0_)/dt], where Kd is the affinity constant of YFP for I^−^ [27,29].

### 2.11. Active RAC Pull-Down Assay

The active RAC pull-down assay was performed as described previously [25,30], using the CRIB (CDC42/RAC1 Interactive Binding) domain of the CDC42/RAC1 effector protein, p21-activated kinase 1 (PAK1) that specifically binds to the active, GTP-bound form of the RAC1 protein. In brief, after the indicated treatments, the cells were washed in cold PBS and lysed on ice in lysis buffer [50 mM Tris–HCl pH 7.5, 100 mM NaCl, 1% (*v*/*v*) NP-40, 10% (*v*/*v*) glycerol, 10 mM MgCl_2_, and a protease inhibitor cocktail (Sigma-Aldrich, St. Louis, MO, USA)]. The lysates were cleared by centrifugation at 2500× *g* for 5 min, and an aliquot of 40 μL (input) was solubilized in Laemmli sample buffer. The remaining lysate was incubated for 1 h at 4 °C with a biotinylated CRIB-domain peptide pre-coupled to streptavidin-agarose beads. The beads were washed three times with lysis buffer, and the captured fractions were solubilized in Laemmli sample buffer. The samples were analyzed by Western blot (see above).

### 2.12. TCGA Data Extraction and General Statistical Analysis

Data from the Thyroid Cancer Dataset at The Cancer Genome Atlas (GDC TCGA THCA) was retrieved using the UCSC Xena online tool [31]. All of the statistical analysis was performed using GraphPad Prism 5 software. The quantitative results are shown as means ± standard error of the mean (SEM) from at least three independent experiments. Student T tests were used for the paired comparisons, and one-way ANOVA followed by post-hoc Tukey HSD or Dunnett’s test were used for multiple comparisons. In all of the analyses, the significance level was set at 0.05, and the *p*-values are as follows: *p* ≤ 0.05 (*), *p* ≤ 0.01 (**) and *p* ≤ 0.001 (***).

## 3. Results

### 3.1. Production and Functional Validation of the Stable HA-NIS-Expressing TPC1 Cell Line

With the aim of selectively studying NIS posttranslational regulation, particularly regarding the regulation of NIS PM abundance, retention and function in vitro, we developed a novel cellular tool. We used the well-characterized TPC1 cell line [32,33], which was derived from a papillary thyroid carcinoma that does not express NIS endogenously [34]. These cells were modified to stably express a functional full-length NIS construct that is constitutively expressed from a CMV promoter and contains a triple HA-epitope tag in the fourth extracellular loop (the largest of the non-glycosylated NIS loops) (Figure 1A), enabling the selective analysis of NIS proteins at the PM of these cells.

For the functional characterization of the HA-NIS-TPC1 cell tool, the total and plasma membrane expression of the HA-NIS protein were first evaluated by Western Blot, in comparison with ectopically expressed full length WT-NIS. The total HA-NIS expression was confirmed in permeabilized cells using both anti-HA and anti-NIS antibodies. Both WT- and HA-tagged proteins produced equivalent size bands, indicative of similar processing through the exocytic pathway (Figure 1B). Moreover, in intact unpermeabilized cells, the extracellular HA-tag allowed the specific detection of the HA-NIS protein at the PM by either immunofluorescence labeling (Figure 1C) or using a surface protein biotinylation assay (Figure 1B). Notably, equivalent levels of HA-NIS and WT-NIS total protein produced similar levels of these proteins at the cell surface (Figure 1B), indicating that the tagged protein has equivalent trafficking to WT-NIS. These results thus confirmed that the HA-NIS protein was correctly produced and trafficked to the cell surface. Second, the NIS function was assessed in the TPC1 cell line by the co-expression of HA-NIS or WT-NIS, together with the halide-sensitive mutant of yellow fluorescent protein (YFP) F46L/H148Q/I152L (HS-YFP), which is quenched by iodide and has been successfully employed to study iodide transport in various cellular systems [27,29]. Using a live cell imaging-based iodide influx assay [24] (Figure 1D), we observed a significant and, again, equivalent decrease in HS-YFP fluorescence over time (Figure 1E) in TPC1 cells expressing either HA-NIS or WT-NIS, which is indicative of an effective rate of iodide uptake into TPC1 cells by the tagged protein at the PM (Figure 1F). We further confirmed that this iodide uptake was dependent on HA-NIS activity, as the effect was abolished upon incubation with perchlorate (ClO_4_^−^), a specific inhibitor of iodide uptake by NIS proteins [35,36].

### 3.2. NIS PM Interactome Is Enriched in Proteins Associated with Actin Cytoskeleton Dynamics

The extracellular HA-tag permits the selective immunolabelling of NIS proteins at the PM, and was employed to identify the signaling molecules and pathways involved in posttranslational regulation of NIS abundance and function. First, in order to selectively isolate NIS-containing multi-protein complexes at the PM, intact HA-NIS-TPC1 cells were incubated on ice with an immunoprecipitation (IP)-grade anti-HA antibody. After thorough washing, the cells were lysed in mild, non-denaturing conditions, and the antibody-labelled NIS complexes from the PM were precipitated with protein G magnetic beads. In parallel, HA-NIS-containing protein complexes were also isolated from whole cell (WC) lysates using a conventional co-immunoprecipitation protocol with the same anti-HA antibody (Figure 2A), in order to subsequently enable the identification of the proteins interacting with NIS exclusively in PM-associated complexes (see below and Figure 2C). A non-specific IgG was used as a control in both approaches to identify potential background contaminants. In order to control for equivalent yields between the different IP conditions or experimental replicates, 20% of the IP eluates were analyzed by Western blot to visualize the HA-NIS protein (Figure 2B). Three experimental replicates of the PM and WCL NIS co-precipitates were sent for analysis by mass spectrometry (MS).

Considering all of the experimental replicates, a total of 384 and 136 proteins were identified with confidence scores (Cf) above 95% in, respectively, the NIS PM-IP and NIS WC-IP samples. This difference in the number of proteins identified in the two datasets would suggest that the localization at the PM requires, or enables, NIS to establish protein–protein interactions that are more resistant to lysis and immunoprecipitation procedures than those established during prior steps of its maturation and trafficking within the cell. However, one must also consider that, at least in part, this difference may also reflect a higher level of contaminants in the PM IPs due to the larger amount of cells required to isolate robust amounts of NIS protein from the PM. Indeed, a larger number of background proteins was also detected in the respective control IPs-136 proteins compared to the 88 proteins detected in the control PM-IP and WC-IPs, respectively. Thus, to ensure the selection of only high-confidence candidate NIS interactors among the WC- and PM-IP datasets, we employed an algorithm that only considered as reliable candidates proteins that met all the following criteria: (1) they were detected in at least two experimental replicates; (2) they were absent in the control IP, or were detected in no more than one control IP, but then the average Cf value in the NIS-IP replicates had to be higher than that of the control IP. This analysis highlighted 52 and 128 high-confidence candidate NIS interactors form the WC-IP and PM-IP datasets, respectively, of which 19 were common to both datasets (Figure 2C, and see Appendix A).

A gene ontology (GO) analysis of the candidate short lists, using the STRING algorithm (https://string-db.org/, accessed on 3 May), revealed significant enrichment in the proteins associated with transport and the secretory pathway among the 33 candidates exclusively detected in the NIS-WC dataset (Table 1 and Appendix A). Surprisingly, whereas transport-associated proteins also accounted for 14 of the 19 candidates common to the NIS-WC and NIS-PM datasets (Table 1 and Appendix A), among the NIS-PM-exclusive candidates, the top five most-represented processes related to the organization of the actin cytoskeleton (Table 1 and Appendix A). Moreover, the KEGG analysis revealed that the “Regulation of actin cytoskeleton” pathway (hsa04810) was highly overrepresented in this dataset (FDR = 1.82 × 10^−^^5^). Further analysis with STRING software revealed that the 29 top-scoring NIS-PM candidates formed a tight interaction subnetwork around cytoplasmic actin (ACTB) and four of the regulatory proteins on the hsa04810 pathway (Figure 3A): the small GTPase RAC1 (RAS-related C3 botulinum toxin substrate 1), the proto-oncogene tyrosine-protein kinase Src (SRC), the actin-nucleating ARP 2/3 complex subunit 4 (ARPC4), and the actin-binding adaptor protein ezrin (EZRIN).

In order to validate the predicted candidate proteins from the complex, we repeated the PM- and WC-NIS IPs, as before, and analyzed the co-precipitating proteins by Western blot (WB), using specific antibodies. We verified that the core PM-NIS candidates, RAC1 and EZRIN, were detected only co-precipitating with NIS isolated from the PM, whereas COPA (coatomer subunit alpha), a protein with a major role in vesicle-mediated transport [37], that was present only in the WC-NIS dataset (see Appendix A and Appendix A), was detectable solely in coprecipitates of NIS isolated from WC lysates (Figure 3B). In addition, IQGAP1 (RAS GTPase-activating-like protein IQGAP1), a scaffold protein known to integrate signals regulating cell adhesion, cell cycle, and the actin cytoskeleton [38], which was common to the PM- and WC-NIS MS datasets (see Appendix A), was readily detected co-precipitating with NIS isolated either from WC lysates or selectively from the PM (Figure 3B). As an additional specificity control, all of the WB were also probed for PCNA (proliferating cell nuclear antigen), a predominantly nuclear protein that was absent from both the PM- and WC-NIS datasets, and, consistently, was also undetectable in either the WC- or PM-NIS co-precipitates (Figure 3B). These results support the robustness of both the MS data and the subsequent bioinformatic analysis.

### 3.3. RAC1 Upregulates NIS Plasma Membrane Levels by Promoting Actin Polymerization and EZRIN Recruitment

We then proceeded to test whether these validated proteins affected NIS trafficking and PM abundance. First, we downregulated the endogenous levels of each of the four candidate NIS interactors using commercial, validated, short, interfering RNAs (siRNAs) (Figure 4A). As expected, a 64 ± 6% depletion of COPA, a major regulator of transport between the endoplasmic reticulum and Golgi in the secretory pathway [37,39,40,41], reduced the amount of NIS delivered to the PM to less than one third of that in the control cells (siCtrl) (Figure 4A,B). In contrast, IQGAP1 downregulation (to 57 ± 3% of siCtrl) did not appear to significantly affect the NIS PM abundance or its overall levels (Figure 4A,B), suggesting that the presence of IQGAP1 in both the PM and WC precipitates is not functionally important for NIS trafficking or membrane retention. Importantly, the depletion of either EZRIN or RAC1 (to 41 ± 9% and 57 ± 2% of siCtrl, respectively) had a clear and significant impact on the NIS abundance at the PM, nearly halving the symporter’s levels at the cell surface without affecting its overall abundance (Figure 4A,B). Because RAC1 is a GTPase—a class of proteins that act as molecular switches by alternating between an active, GTP-bound state and an inactive, GDP-bound state—the inhibition of its activation state should be sufficient to interfere with its cellular role [42]. Consistently, when we treated HA-NIS-TPC1 cells with EHT1864, a specific inhibitor of RAC1 activation (see Appendix A), the reduction in the NIS PM levels became even more significant, decreasing to nearly one third of the control (Figure 4C).

Notably, both the depletion of EZRIN and the treatment with EHT1864 significantly decreased the NIS-dependent iodide influx in HA-NIS-TPC1 cells (Figure 4D), in agreement with the observed reduction in NIS PM levels. In addition, in order to ascertain that the observed effects were not exclusive to the cell line model used, we inhibited the endogenous RAC1 signaling in the non-transformed, follicular thyroid cell line PCCL3, and analyzed the effects on the surface abundance of endogenous NIS. PCCL3 cells respond to TSH stimulation by notably increasing the expression and surface abundance of endogenous NIS protein (Figure 4E). However, upon a short 1 h treatment with EHT1864, sufficient to inhibit the endogenous RAC1 activity in these cells (see Appendix A), the NIS levels at the PM were significantly decreased, while the total levels in the WCL remained unchanged (Figure 4E). Moreover, treatment with EHT1864 also produced a consistent decrease in the iodide uptake by TSH-stimulated PCCL3 cells stably expressing the HS-YFP sensor (Figure 4F).

Conversely, the transfection of HA-NIS-TPC1 cells with constitutively active RAC1 mutants (RAC1-L61 or RAC1-V12; see also Appendix A) or ectopic EZRIN, had the opposite effect, i.e., all nearly doubled the NIS abundance at the PM (Figure 5A,B) and also consistently increased the NIS-mediated iodide influx into HA-NIS-TPC1 cells (Figure 5C).

RAC1 signaling is known to induce EZRIN recruitment and binding to the cortical actin cytoskeleton [27,43]. Thus, we investigated whether the effects of RAC1 and EZRIN on the NIS PM levels were part of the same pathway. We observed that the increased NIS PM levels upon expressing constitutively active RAC1 was reverted when EZRIN was concomitantly knocked-down (Figure 6A). Moreover, the increment in the NIS-PM observed upon EZRIN overexpression was completely abolished when the cells were co-treated with EHT1864 (Figure 6B). These data strongly suggest that RAC1 activity is required upstream of EZRIN to promote NIS PM residency.

EZRIN is an actin-binding adaptor protein that anchors PDZ-containing membrane proteins (like NIS) to the actin cytoskeleton [44,45,46], whereas RAC1 is as a major regulator of de novo filamentous (F-) actin branching and extension, providing anchoring points for EZRIN at the PM [47,48,49,50,51]. RAC1-induced F-actin polymerization is largely mediated by the ARP2/3 complex, a seven-subunit assembly that functions as an actin nucleator, extending actin filaments into branched networks [50,51,52,53]. Because our MS analysis revealed that both actin and ARP2/3 complex subunits co-precipitated with NIS from the PM, we investigated whether ARP2/3-mediated actin polymerization was required for the RAC1-induced upregulation of NIS at the cell surface. We observed that interference with ARP2/3 activity, either through treatment with the chemical inhibitor CK666 or by knocking down (31 ± 7% of siCtrl) the actin nucleating subunit ARP3 (siARP3), significantly decreased the NIS PM abundance in HA-NIS-TPC1 cells (Figure 6C). Moreover, the downregulation of ARP2/3 completely prevented a constitutively active RAC1 mutant from increasing the NIS PM levels (Figure 6D), as is consistent with the requirement of RAC1-induced de novo actin polymerization to allow the retention of NIS at the PM.

### 3.4. RAC1 Signals through PAK1 and PIP5K Kinases to Upregulate NIS PM Levels

RAC1, together with RHOA and CDC42, are the canonical members of the Rho family of small GTPases, all of which participate in the regulation of actin cytoskeleton dynamics [48,49,54,55,56,57]. In order to ascertain that the observed effect on NIS PM residency was specific to RAC1 signaling, we transfected HA-NIS-TPC1 cells with constitutively active or dominant negative mutants of RHOA (RHOA-L63 or -N19, respectively) and CDC42 (CDC42-L61 or -N17, respectively) known to, respectively, stimulate or inhibit signaling by their endogenous counterparts [28]. We observed that, in contrast to RAC1, neither the active nor the dominant negative mutants of the other two canonical GTPases affected NIS PM abundance (Appendix A).

Once it was confirmed that NIS PM abundance was specifically controlled by RAC1 signaling, we investigated which of the RAC1 direct effectors could be mediating the observed effects. None of the known RAC1 direct effectors, including downstream protein kinases, were present among the MS-detected PM-NIS candidate interactors, suggesting that their transient association with the GTPase could elude co-capture with the PM-NIS complexes. P38 mitogen-activated protein kinase and c-Jun N-terminal kinase (JNK) are two known RAC1 effectors that have previously been implicated in the regulation of NIS expression at the transcription level [58,59,60]. However, treatment with the selective inhibitors of JNK and p38, SP600125 or SB303580, respectively, induced no significant effect on the NIS PM abundance in HA-NIS-TPC1 cells (Appendix A).

Another well-known RAC1 effector is p21-activated kinase 1 (PAK1), a serine/threonine protein kinase that has been implicated in both RAC1-induced F-actin polymerization [61,62,63] and EZRIN activation [64,65], two RAC1-induced events which have been shown above to promote NIS residency at the PM. Consistently, an 83 ± 8% depletion of PAK1 was able to induce a nearly 2-fold decrease in PM NIS levels, and this effect was equivalent to that in cells treated with the PAK1 selective inhibitor IPA3 (Figure 7A). Importantly, PAK1 downregulation reverted the increase in NIS PM abundance induced by constitutively active RAC1 expression (Figure 7B), indicating PAK1 as a key mediator of RAC1-induced NIS upregulation at the PM. At the functional level, both the siRNA-mediated and chemical inhibition of PAK1 produced a strong effect on the cellular iodide uptake activity, reducing the influx rates more than twofold (Figure 7C).

Phosphatidylinositol-4-phosphate-5-kinase (PIP5K) is also a known RAC1 effector that facilitates PAK1 activation and RAC1-induced actin polymerization [66,67], and mediates the activation of EZRIN by RAC1 at the PM [68]. Consistently, a 76 ± 8% siRNA-mediated downregulation of PIP5K was sufficient to reduce by half the NIS abundance at the PM (Figure 8A), as well as the corresponding iodide uptake by HA-NIS-TPC1 cells (Figure 8B). Moreover, PIP5K downregulation also prevented constitutively active RAC1 from increasing the NIS PM levels (Figure 8C), which is in agreement with there being a pivotal role for PIP5K in the RAC1-induced retention of NIS at the PM.

### 3.5. SRC Kinase Functions Upstream of RAC1 to Increase the NIS Cell Surface Levels

SRC kinase was among the core PM-NIS interactors involved in the regulation of the actin cytoskeleton (see Figure 3A). The SRC family of tyrosine kinases regulates a broad spectrum of actin cytoskeleton-related cellular events, ranging from cell division to cell adhesion and motility [69]. SRC acts downstream of many receptor types and signals through a myriad of pathways, many of which lead to the activation of RAC1 [70]. In accordance with this, when we treated HA-NIS-TPC1 cells with SRC inhibitors PP2 or Dasatinib, a clear, over-twofold reduction in the endogenous active RAC1 (GTP-bound) levels was observable (Figure 9A). Notably, this downregulation of RAC1 activity by SRC inhibitors was accompanied by a reduction of the same magnitude in the NIS PM levels (Figure 9B), which in turn was reflected in a significant decrease in the NIS-mediated iodide uptake (Figure 9C). Taken together, these observations indicate that SRC is responsible for relaying the extracellular signals that induce the activation of RAC1 in the NIS-multi-protein complex at the PM. Again, in order to ascertain that the observed effects were not exclusive to TPC1 cells, we chemically inhibited endogenous SRC signaling with PP2 in TSH-stimulated PCCL3 expressing the HS-YFP sensor, and verified a significant decrease in iodide uptake (Figure 9D), as is consistent with the decrease observed upon RAC1 inhibition (see Figure 4F). In addition, the chemical inhibition of the key components of the characterized pathway downstream of RAC1, namely PAK1 and ARP2/3, also resulted in an equivalent impairment of the iodide uptake in these cells. These observations suggest that the same pathway may act to promote the PM residency of endogenous NIS in PCCL3 cells, further supporting the observations made in TPC1 cells.

## 4. Discussion

Differentiated thyroid tumors often exhibit reduced, or sometimes undetectable, I^−^ transport compared with normal thyroid tissue, and they are diagnosed as cold nodules on thyroid scintigraphy [71]. Despite this reduction, the majority of DTCs accumulate sufficient I^−^ levels to enable RAI treatment [9,71]. Unfortunately, 30% to 50% of the metastases from differentiated thyroid tumors completely lose their ability to accumulate I^−^, causing them to become refractory to RAI therapy [10,71].

Importantly, immunohistochemistry studies revealed that, in many refractory metastatic papillary carcinomas, the reduction in the I^−^ uptake involves impaired NIS transport and residence at the plasma [18,72]. Indeed, several other studies revealed that some cold thyroid nodules express normal, or even higher, NIS levels compared with adjacent normal tissue, but NIS is frequently intracellularly retained, suggesting the presence of posttranslational abnormalities in the transport and retention of the protein at the PM [8,18,19,73,74,75,76,77]. These data highlight the pressing need to elucidate the posttranslational mechanisms that regulate NIS expression at the PM, both under physiological and pathological conditions, so as to devise new strategies to restore the RAI uptake by refractory thyroid carcinomas.

Despite the physiological and clinical relevance of NIS PM expression in thyroid carcinomas, little is known about the molecular mechanisms underlying the NIS transport to, and retention at, the PM [7]. A few studies identified affected membrane trafficking steps. PI3K-AKT pathway activation was reported to downregulate the NIS glycosylation and surface translocation in non-transformed rat thyroid cells or human papillary thyroid carcinoma cell lines [78]. The decreased expression of the ribosomal machinery subunit phosphatidylinositol glycan anchor biosynthesis class U(PIGU), downstream of upregulated MEK activity, resulted in improper NIS posttranslational processing and deregulated trafficking to the PM in DTC cell lines, and was associated with a decreased response to RAI treatment in patients with recurrent DTC [79]. Pituitary tumor transforming gene 1 (PTTG1) and PTTG-1 binding factor (PBF) overexpression in thyroid cancers can decrease the NIS protein levels [80], possibly by repressing NIS mRNA transcription or by removing the protein from the PM [20]. Indeed, ectopically overexpressed PBF can directly interact with NIS and induce its clathrin-mediated internalization, leading to its intracellular accumulation in late endosomes in thyroid cell lines [20].

In a recent study, Fletcher et al. [23] isolated new putative NIS interactors from MDA-MB-231 breast carcinoma cells ectopically expressing the human NIS protein. Similarly to our whole-cell NIS IP data (WC-NIS), the authors found that the NIS co-precipitated protein complex from MDA-MB-231 cells was also enriched in transport-associated proteins, 13 of which were also present in our WC-NIS dataset (see Appendix A and Appendix A). Among these was COPA, the major subunit of the coatomer protein I (COPI) complex, which operates in the early secretory pathway [37]. Here, we showed that NIS directly interacts with COPA in the cytosol of TPC1 thyroid cells, and that COPA downregulation impairs NIS trafficking to the PM, but additionally seemed to interfere with the symporter’s synthesis or processing, as the NIS overall levels also decreased. Fletcher et al. [23] identified ADP-ribosylation factor 4 (ARF4) and valosin-containing protein (VCP) as two additional NIS binding partners that antagonistically regulate NIS trafficking to the PM: ARF4 facilitates the Golgi-PM trafficking of mature NIS proteins, whereas VCP, an ER-associated protein, acts similarly to PBF, and decreases NIS-mediated iodide uptake by promoting the ER-associated proteasomal degradation of immature NIS proteins [23]. Although ARF4 and VCP were not among the proteins detected with high confidence in our PM-NIS dataset, possibly due to the different cellular systems and methodological variations, we did find 14 proteins in common, most of which were associated with ER-Golgi and Golgi-PM trafficking (see Appendix A). Notably, our study extends the study by Fletcher et al. by providing a detailed analysis of the proteins interacting with NIS exclusively at the PM. This analysis revealed that although 28% of these proteins were associated with vesicle mediated transport, a similar percentage (27%) was associated with the regulation of the actin cytoskeleton (see Table 1 and Appendix A). Moreover, the enrichment in the proteins of the latter category was nearly five orders of magnitude more significant (4.51 × 10^−11^ vs. 1.82 × 10^−6^) in our PM-NIS dataset. These observations suggested that the actin cytoskeleton and its regulatory machinery could play an important role in the mechanisms enabling NIS prevalence at the PM. In agreement with this hypothesis, we found that in the core of this regulatory network, alongside actin, was the small GTPase RAC1. We and others have reported that RAC1 signaling potentiates the TSH-induced transcription of the NIS (SLC5A5) gene in non-transformed thyroid cells [24], and also in MCF7 breast cancer cells, in this case through the activation of p38 MAPK signaling [59]. Here, we found that RAC1 signaling also has an important role in the posttranscriptional regulation of NIS PM residency. For this, RAC1 signals neither through p38 nor through JNK (another kinase downstream of RAC1 known to participate in the transcriptional regulation of NIS [60]), but through PAK1 and PIP5K. The combined activity of these kinases induces ARP2/3-mediated actin polymerization, and the recruitment and activation of EZRIN, which binds NIS and increases its membrane residency, likely by anchoring it to the cytoskeleton and promoting its retention at the cell surface (see diagram in Figure 10). EZRIN interacts with membrane proteins either directly or indirectly, through postsynaptic density 95/disc-large/zona occludens (PDZ) domain-containing scaffold proteins of the Na^+^/H^+^ exchanger regulator (NHERF) family [45].

Although NIS was described to contain an internal and a C-terminal PDZ-binding domain [81], our MS analysis detected no NHERF family scaffold protein co-precipitating with NIS from the PM (see Appendix A). Moreover, the re-probing of the PM co-IP WB membranes used to validate the EZRIN-NIS interaction with anti-NHERF-1 antibodies failed to detect the presence of this scaffold protein (see Appendix A). Although the participation of a PDZ-binding protein in NIS-EZRIN complex formation cannot be definitely excluded, these observations suggest that EZRIN might bind directly to NIS, possibly through an as yet unidentified FERM-binding domain in the symporter. While FERM-binding domains have highly heterogeneous sequences, Ali et al. [82] recently characterized an xYxV motif that is present in many of EZRIN interacting partners, and mediates the interaction with the F3b site in its FERM domain. Interestingly, by analyzing the NIS protein sequence, we detected two putative FERM-interacting xYxV motifs near the interfaces between the second NIS cytoplasmic loop and the fourth transmembrane domain, and between the seventh NIS transmembrane domain and fourth cytoplasmic loop (see Appendix A). Whether or not these regions mediate the NIS-EZRIN interaction downstream of RAC1 will require further investigation.

Another interesting finding in this work was that SRC kinase acts upstream of RAC1 in promoting NIS PM residency. It has been suggested that the pharmacological inhibition of SRC signaling could be beneficial in the treatment of advanced thyroid cancer, as it was shown to reduce tumor cell growth and metastasis, both in vitro and in vivo [83,84,85]. However, SRC inhibitors have had limited efficacy in clinical trials for solid tumors, likely due to resistance mechanisms [86,87,88]. Moreover, recent evidence suggests that the acquisition of resistance to SRC inhibition in BRAF-mutant thyroid cancer cells increases the expression of pro-inflammatory cytokines and pro-invasive metalloproteases, promoting a more invasive phenotype [89]. Our findings add a new layer of complexity, as they indicate that the administration of SRC inhibitors could compromise therapeutic strategies envisioning the re-sensitization of RAI-refractory DTCs through the increase of NIS expression, as the inhibition of SRC may concurrently hamper RAC1 activation. This becomes even more relevant in light of promising results recently obtained in patients after the mutation-guided redifferentiation of radioiodine refractory DTCs via MAPK inhibition [71,90], in which the use of an MEK inhibitor (in patients with mutant NRAS) or a combination of BRAF and MEK inhibitors (in patients with mutant BRAF) partially restored the RAI uptake in three of three BRAF-positive patients, and one of three NRAS-positive patients, enabling RAI therapy [90]. Curiously, the analysis of the thyroid cancer dataset at The Cancer Genome Atlas (GDC TCGA THCA) revealed positive, significant correlations among the RAI-treated patients, between the total administered dose of I^131^, SRC expression in the primary tumor (r = 0.19; *p* = 0.0124), and patient overall survival (OS) (r = 0.15; *p* = 0.0415) (Appendix A). Although the identified associations would require careful validation, if one would speculate that a larger I^131^ dose is suggestive of RAI-responsiveness (hence the better OS), then higher SRC levels could favor iodide uptake in primary tumors, as is consistent with better NIS functionality at the PM.

These observations highlight the need to further investigate which upstream events trigger the identified SRC/RAC1/PIP5K/PAK/EZRIN/NIS pathway in normal thyroid cells. The SRC kinase family regulates a broad spectrum of cellular processes, including cell proliferation, cell adhesion, and cell motility [69,88], downstream of a myriad of receptors, many of which signal towards RAC1 activation [70]. The future clarification of which events are upstream of this newly identified pathway may thus provide new therapeutic possibilities, namely the modulation of the SRC activity in refractory TC, to switch it from promoting proliferation and invasion to enhancing the NIS retention at the PM, contributing to the re-sensitization of these tumors to RAI therapy.

## Figures and Tables

**Figure 1 cancers-13-05460-f001:**
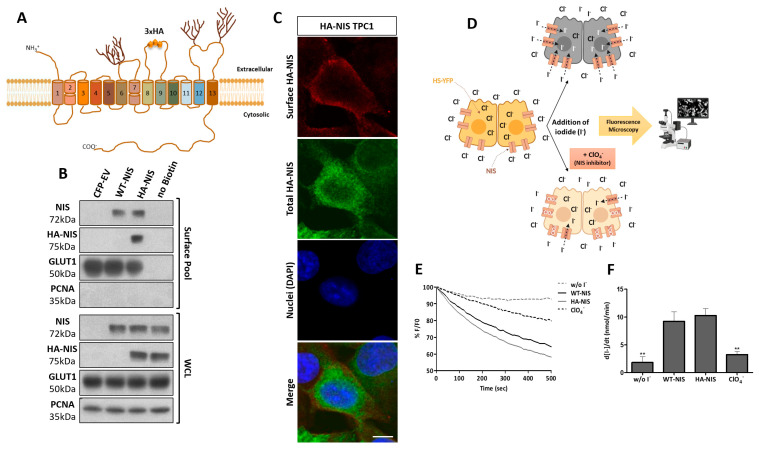
Production and characterization of the HA-NIS-expressing TPC1 cell line. (**A**) Diagram depicting the topology of NIS protein, showing the 13 transmembrane domains and the location of the inserted 3 × HA tag at the fourth extracellular loop. The TPC1 cells were modified to express the HA-NIS fusion protein, and then the cell line model was characterized regarding HA-NIS protein expression, trafficking to cell membrane and iodide transport function. (**B**) Cell surface proteins were biotinylated, and both the whole-cell lysates (WCL) and surface protein fraction were analyzed by Western blot to detect the HA-NIS protein, using anti-HA and anti-NIS primary antibodies. TPC1 expressing either CFP or WT-NIS cells was used as the negative and positive control, respectively, for the HA-NIS protein expression. The ‘no Biotin’ condition, corresponding to cells that were not incubated with biotin, was used as the control for non-biotinylated protein capture. PCNA served as the loading (WCL) and intracellular protein contamination control (Surface pool). GLUT-1 served as the positive loading control for the cell-surface protein extracts. (**C**) Confocal images of tripled labelled HA-NIS-TPC1 cells, where Alexa Fluor 633 fluorescence (red) indicates HA-NIS protein present at the surface of intact cells, labeled with a rabbit anti-HA primary antibody, prior to the cell fixation and permeabilization. Alexa Fluor 488 fluorescence (green) indicates the total HA-NIS protein, labeled with a mouse anti-HA primary antibody after cell permeabilization. The nuclei were stained with DAPI (blue). (**D**) Diagram depicting the fluorescent iodide influx assay. HA-NIS-TPC1 cells stably expressing the YFP halide sensor (HS-YFP) were used to analyze the HA-NIS protein function. The decay of the YFP fluorescence was monitored continuously for 500 s, acquiring an image every 10 s, after exposing the cells to 50 mM NaI. The presence of a competitive inhibitor of iodide uptake by NIS (ClO_4_^−^) slows or prevents the decay of YFP fluorescence over time (image created with BioRender.com, accessed on 26 July 2021). (**E**) Representative traces of HS-YFP decay upon the exposure of WT-NIS- or HA-NIS- expressing TPC1 cells to iodide (I^−^), in the presence or absence of pretreatment with 50 mM ClO_4_^−^. Fluorescence [F] was plotted over time as percentage of the fluorescence at time 0 [F_0_]. “w/o I^−^” relays the fluorescence values in cells assayed in I^−^-free medium. (**F**) Iodide influx rates calculated by fitting the curves to the exponential decay function to derive the maximal slope that are proportional to the initial rate of I- influx into the cells. The data are means ± SEM of five independent assays. A one-way ANOVA analysis detected significant differences between the treatments (F = 15.02; *p* ˂ 0.001). Post-hoc Dunnett’s tests were used to identify significant variations relative to WT-NIS (** *p* ≤ 0. 01).

**Figure 2 cancers-13-05460-f002:**
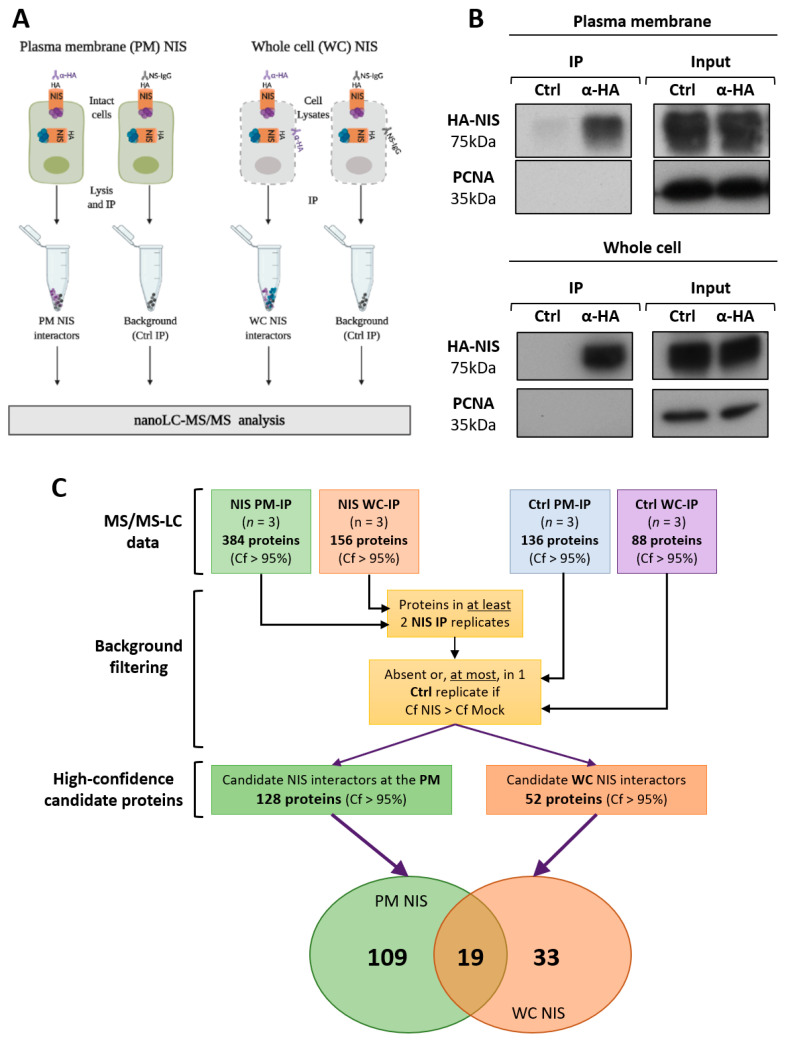
Sample preparation for the MS analysis. (**A**) Schematic representation of the protocols used to precipitate HA-NIS-containing complexes from the PM of intact cells, and to precipitate HA-NIS protein complexes from whole cell (WC) lysates (figure created with BioRender.com, accessed on 26 July 2021). Both WC- and PM-HA-NIS-containing complexes were selectively immunolabeled using a rabbit anti-HA antibody. Non-specific IgG (Ctrl) was used to control IP contaminants. (**B**) WC and PM immunoprecipitates were analyzed by Western blot to confirm IP robustness. The total (input) and IPed HA-NIS proteins were detected using a mouse anti-HA primary antibody. PCNA expression was used as the loading (input) and contamination control (IP). (**C**) Schematic representation of the pipeline used for the MS data analysis (see the text for details).

**Figure 3 cancers-13-05460-f003:**
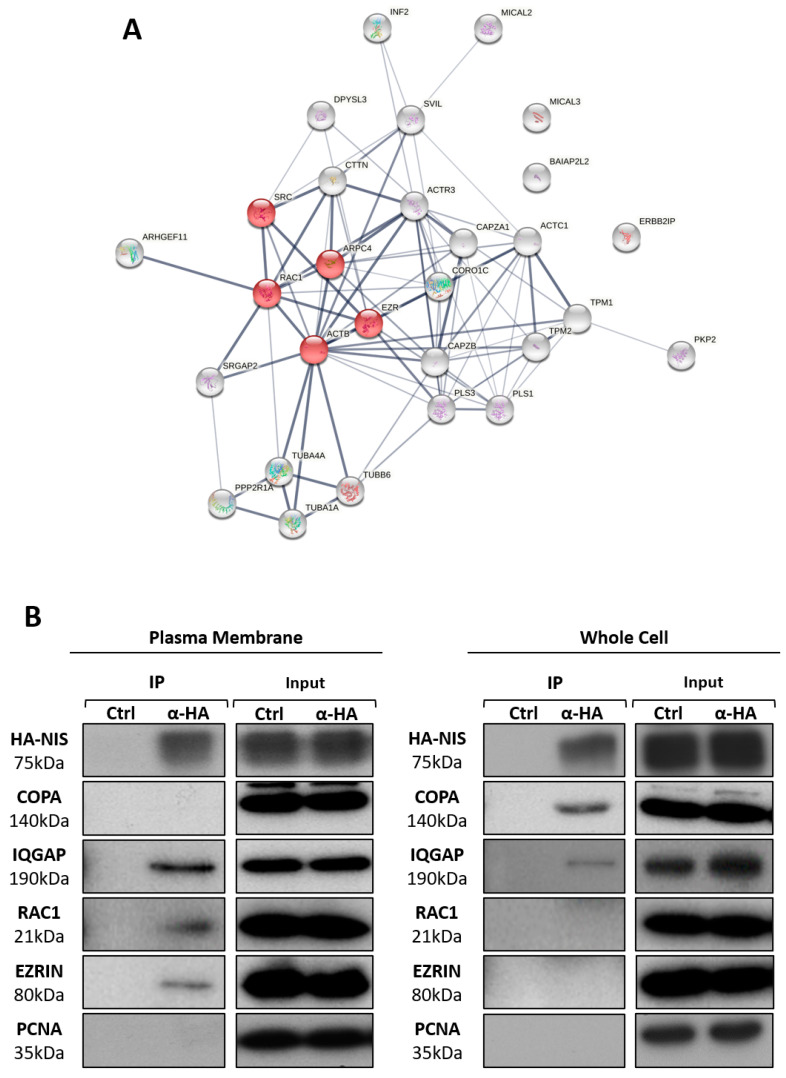
Validation of high-confidence NIS interactions detected by MS analysis. (**A**) STRING-generated subnetwork with the 29 top-scoring NIS-PM candidates. The network nodes represent the identified proteins. The grey lines connecting two nodes represent protein associations extrapolated from textmining-, experimental- and database-collected evidence. The thickness of the lines is proportional to the degree of confidence for the predicted association between the nodes. Note that the network forms around cytoplasmic actin (ACTB) and the four other proteins highlighted by KEEG pathway analysis (in red) as involved in “Regulation of actin cytoskeleton” (hsa04810). (**B**) NIS-WC and NIS-PM immunoprecipitated protein complexes were analyzed by Western blot, using specific primary antibodies to detect the indicated target candidate proteins. The total (input) and IP HA-NIS levels were detected using a mouse anti-HA primary antibody to confirm the immunoprecipitation efficiency.

**Figure 4 cancers-13-05460-f004:**
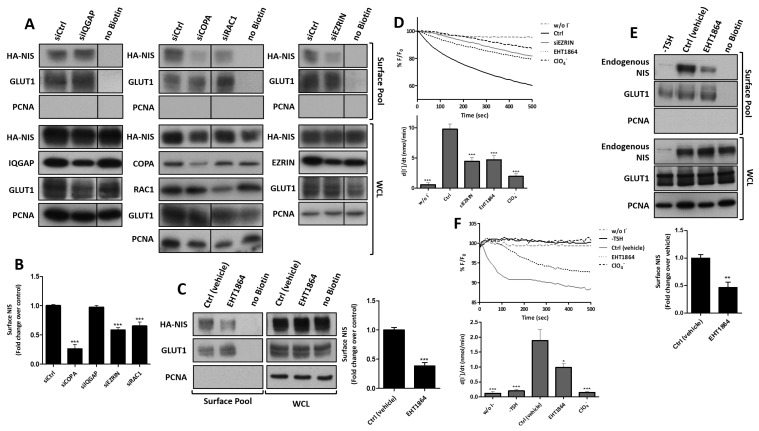
Effects of IQGAP, COPA, RAC1 and EZRIN depletion on the NIS abundance at the PM. (**A**) HA-NIS-TPC1 cells were transfected with siRNAs targeting IQGAP, COPA, EZRIN and RAC1, and were analyzed by cell-surface protein biotinylation. The HA-NIS protein in either the surface fraction or the correspondent whole-cell lysates (WCL) was detected by WB. The ‘no Biotin’ condition, corresponding to cells that were not incubated with biotin, was used to control the specificity of the biotinylated protein pull-down. WCL were further probed for IQGAP, COPA, RAC1 and EZRIN protein levels, in order to ascertain siRNA efficiency. PCNA detection served as the loading (WCL) and intracellular protein contamination control (Surface pool). GLUT-1 served as the positive loading control for the cell surface protein extracts. (**B**) Surface HA-NIS expression was quantified by the densitometric analysis of the WB bands, using ImageJ software. The plotted values are means ± SEM of three independent assays. The one-way ANOVA analysis detected significant differences between the treatments (F = 33.17; *p* < 0.001). Post-hoc Dunnett’s tests were used to identify significant variations relative to ‘siCtrl’ (transfection with a siRNA against luciferase firefly–see Materials and Methods) (*** *p* ≤ 0.001). (**C**) HA-NIS-TPC1 cells were treated with 50 µM EHT1864 for 1 h and analyzed by cell-surface protein biotinylation followed by WB, as in (**A**). The surface HA-NIS expression was quantified by the densitometric analysis of the WB bands, as before. A Student T test was used to evaluate significant variations relative to the ‘Ctrl’ sample (*** *p* ≤ 0.001). (**D**) HA-NIS-TPC1 cells stably co-expressing the HS-YFP iodide sensor were transfected with either siCtrl or siEZRIN, or treated or not with 50 µM EHT1864 for 1 h, or 50 mM of ClO_4_^−^ for 10 min, and the YFP fluorescence was recorded continuously, as described in the legend to Figure 1. The data are the means ± SEM of five independent assays. One-way ANOVA analysis detected significant differences between the treatments (F = 33.39 and *p* < 0.001). Post-hoc Dunnett’s tests were used to identify significant variations relative to the control (Ctrl) conditions (*** *p* ≤ 0.001). (**E**) PCCL3 thyroid cells were serum-starved for 24 h, followed by stimulation or not with TSH (1 mU/mL for 48 h). The cells were then treated with either vehicle or 50 µM EHT1864 for 1 h, and were analyzed by surface protein biotinylation and WB, as in (**A**). The plotted values are the means ± SEM of three independent assays. Significant variations were assessed by Student T test (** *p* ≤ 0.01). (**F**) PCCL3 cells stably expressing the HS-YFP iodide sensor were stimulated as in (**E**), and were treated or not with 50 µM EHT1864 for 1 h, or 1 mM of ClO_4_^−^ for 10 min, and their YFP fluorescence was recorded continuously, as described in the legend to Figure 1. The data are the means ± SEM of five independent assays. One-way ANOVA analysis detected significant differences between the treatments (F = 18.13 and *p* < 0.001). Post-hoc Dunnett’s tests were used to identify significant variations relative to the control (Ctrl) conditions (* *p* ≤ 0.05; *** *p* ≤ 0.001).

**Figure 5 cancers-13-05460-f005:**
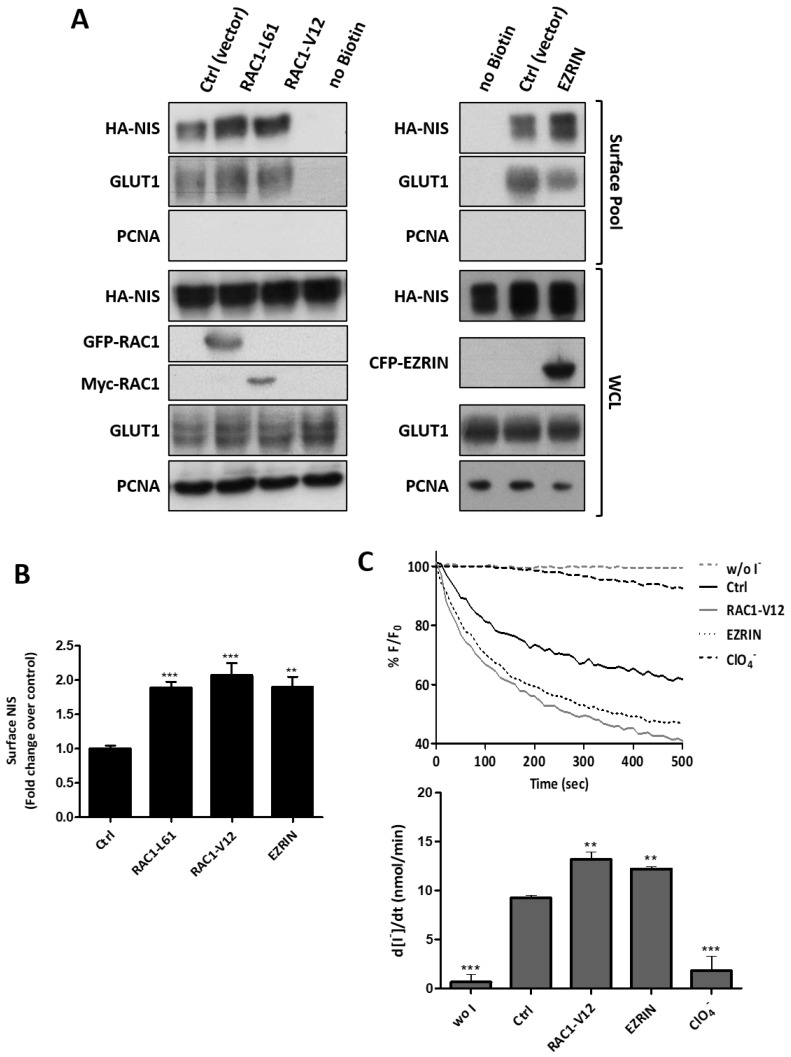
Upregulation of RAC1 and EZRIN increases the NIS residency at the PM. (**A**) HA-NIS-TPC1 cells were transiently transfected with either empty vector (Ctrl), GFP-RAC1-L61, MYC-RAC1-V12 or CFP-ERZIN constructs, and were analyzed by cell surface protein biotinylation. The surface fractions and the corresponding whole-cell lysates (WCL) were analyzed by WB, as indicated. PCNA detection served as the loading (WCL) and intracellular protein contamination control (Surface pool). GLUT-1 served as the positive loading control for the cell-surface protein extracts. (**B**) WB bands were quantified by densitometric analysis using ImageJ software. The plotted values are the means ± SEM of five independent assays. A one-way ANOVA analysis detected significant differences between the treatments (F = 17.82; *p* < 0.001). Post-hoc Dunnett’s tests were used to identify significant variations relative to the control (Ctrl) conditions (** *p* ≤ 0.01; *** *p* ≤ 0.001). (**C**) HA-NIS-TPC1 cells stably co-expressing the HS-YFP iodide sensor were transfected with either empty vector (Ctrl), MYC-RAC1-V12 or CFP-ERZIN, or were treated or not with 50 mM of ClO_4_^−^ for 10 min, and YFP fluorescence was recorded continuously (as described in the legend to Figure 1). The data are the means ± SEM of three independent assays. One-way ANOVA analysis detected significant differences between the treatments (F = 59.51 and *p* < 0.001). Post-hoc Dunnett’s tests were used to identify significant variations relative to the control (Ctrl) conditions (** *p* ≤ 0.01; *** *p* ≤ 0.001).

**Figure 6 cancers-13-05460-f006:**
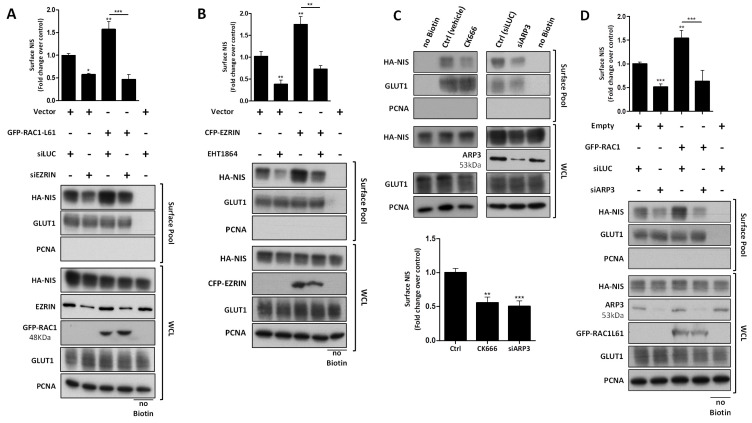
EZRIN and ARP2/3 activity are required downstream of RAC1 to sustain HA-NIS PM residency. HA-NIS-TPC1 cells were transfected with either (**A**) GFP-RAC1-L61 or a siRNA against EZRIN, or both; (**B**) with CFP-EZRIN in the presence or absence of the RAC1 inhibitor EHT1864 (50 µM for 1 h); (**C**) with either mock (siLUC control) or siARP3, and treated or not with the ARP2/3 complex inhibitor CK666 (100 µM for 1 h); or (**D**) with either empty vector (Empty) or GFP-RAC1-L61, and either mock (siLUC control) or the siARP3, as indicated, and analyzed by surface protein biotinylation. Surface fractions and whole-cell lysates (WCL) were then analyzed by WB, as indicated. The ‘no Biotin’ condition corresponds to cells not incubated with biotin, and was used to control the specificity of biotinylated protein pull-downs. PCNA expression served as both the loading (WCL) and intracellular protein contamination control (Surface pool). GLUT-1 served as the positive loading control for the cell-surface protein extracts. The WB bands were quantified by densitometric analysis using ImageJ software. The plotted values correspond to the means ± SEM of at least three independent assays. One-way ANOVA analysis detected the significant differences between the treatments [F = 23.65 and *p* < 0.001 for (**A**); F = 17.86 and *p* < 0.001 for (**B**); F = 14.79 and *p* < 0.001 for (**C**); F = 18.26 and *p* < 0.001 for (**D**)]. Post-hoc Tukey’s tests were used to identify significant variations relative to the control conditions or among the different treatments (the latter are indicated by horizontal lines) (* *p* ≤ 0.05; ** *p* ≤ 0.01; *** *p* ≤ 0.001).

**Figure 7 cancers-13-05460-f007:**
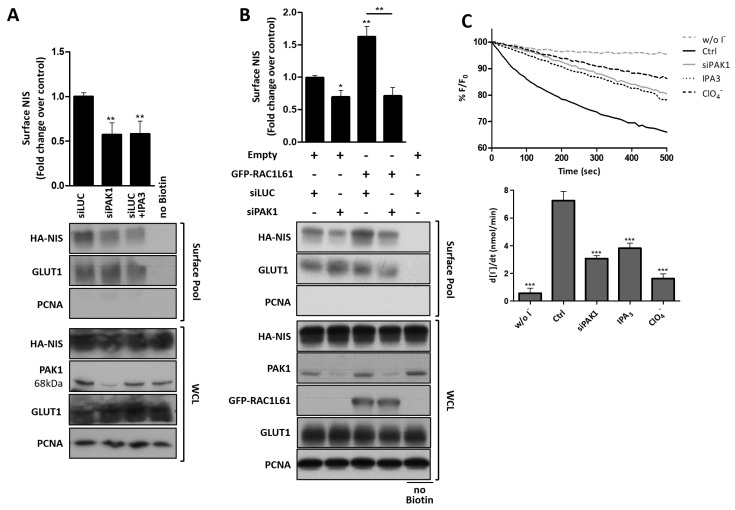
PAK1 participates in the RAC1-stimulated upregulation of NIS PM residency. HA-NIS-TPC1 cells were (**A**) transfected with either mock (siLUC control) or a specific siRNA against PAK1 (siPAK1), or were treated or not with PAK1 chemical inhibitor IPA3 (10 µM for 1 h); or (**B**) transfected with either empty vector (Empty) or GFP-RAC1-L61, and either mock (siLUC control) or the siPAK1, as indicated, and were analyzed by surface protein biotinylation. The surface fractions and whole-cell lysates (WCL) were then analyzed by WB, as indicated. The ‘no Biotin’ condition, which corresponds to cells which were not incubated with biotin, was used to control the specificity of biotinylated protein pull-downs. PCNA expression served both as the loading (WCL) and intracellular protein contamination control (Surface pool). GLUT-1 served as the positive loading control for the cell-surface protein extracts. The WB bands were quantified by densitometric analysis using ImageJ software. The plotted values correspond to the means ± SEM of at least three independent assays. One-way ANOVA analysis detected significant differences between the treatments [F = 9.31 and *p* = 0.0012 for (**A**); F = 12.73 and *p* < 0.001 for (**B**)]. Post-hoc Tukey’s tests were used to identify significant variations relative to the control conditions or among the different treatments (the latter are indicated by horizontal lines) (* *p* ≤ 0.05; ** *p* ≤ 0.01). (**C**) Representative traces of the iodide-induced YFP fluorescence decay of HA-NIS/HS-YFP-TPC1 cells treated as in (**A**), or treated with 50 mM ClO_4_^−^ for 10 min, continuously recorded as described in the legend to Figure 1. The data are the means ± SEM of five independent assays. The significant variations were assessed by one-way ANOVA (F = 30.50; *p* < 0.001) followed by Dunnett’s post- hoc test, as compared with the control conditions (Ctrl) (*** *p* < 0.001).

**Figure 8 cancers-13-05460-f008:**
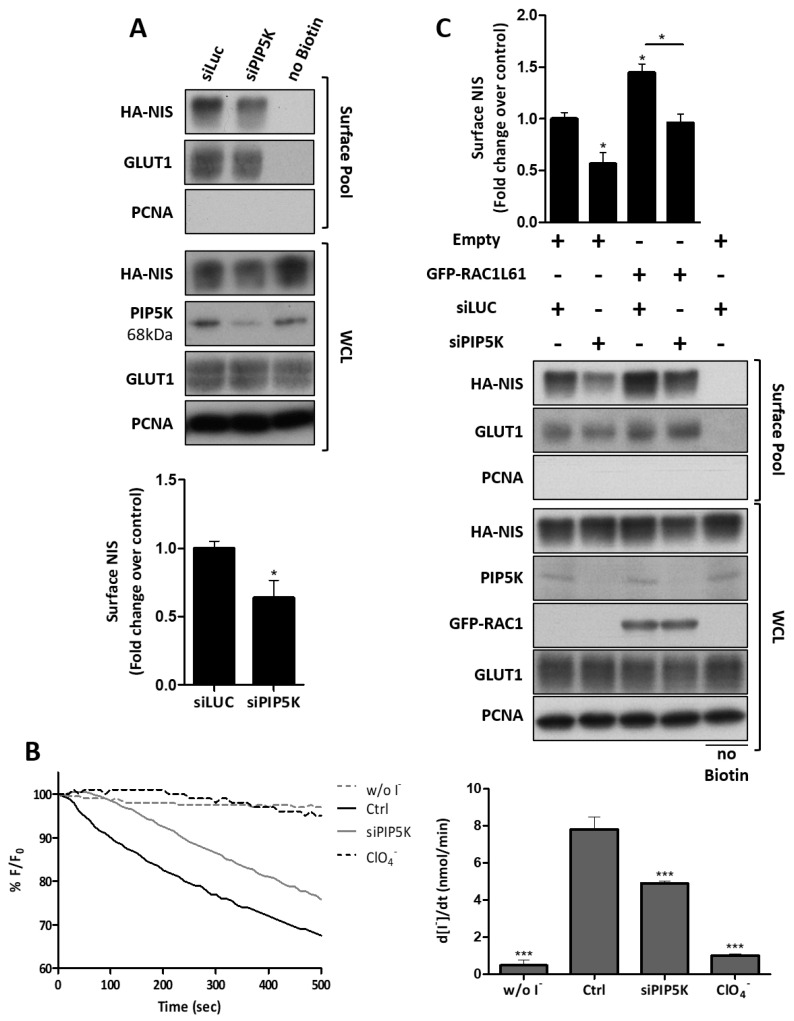
RAC1 signaling through PIP5K is required for NIS PM residency. HA-NIS-TPC1 cells were (**A**) transfected with either mock (siLUC control) or a specific siRNA against PIP5K (siPIP5K), and were analyzed by surface protein biotinylation. The surface fractions and whole-cell lysates (WCL) were then analyzed by WB, as indicated. The ‘no Biotin’ condition, corresponding to cells which were not incubated with biotin, was used to control the specificity of the biotinylated protein pull-downs. PCNA expression served as both the loading (WCL) and intracellular protein contamination control (Surface pool). GLUT-1 served as the positive loading control for cell-surface protein extracts. The WB bands were quantified by densitometric analysis using ImageJ software. The plotted values correspond to the means ± SEM of at least three independent assays. Significant variations were assessed by Student’s T test (* *p* ≤ 0.05). (**B**) Representative traces of the iodide-induced YFP fluorescence decay of HA-NIS/HS-YFP-TPC1 cells treated as in (**A**), or treated with or 50 mM of ClO_4_^−^ for 10 min, which were continuously recorded for 500 s after exposure to 50 mM iodide (upper graph). The iodide influx rates (lower graph) were calculated by fitting the curves to the exponential decay function. The data are means ± SEM of five independent assays. Significant variations between the treatments were assessed by one-way ANOVA (F = 120.8; *p* < 0.001) followed by Dunnett’s post- hoc test, as compared with the control conditions (Ctrl) (*** *p* ≤ 0.001). (**C**) HA-NIS-TPC1 cells transfected with either empty vector (Empty) or GFP-RAC1-L61, and either mock (siLUC control) or the siPIP5K, as indicated, were analyzed by cell surface protein biotinylation followed by WB, as in (**A**). The plotted values correspond to the means ± SEM of at least three independent assays. One-way ANOVA analysis detected significant differences between the treatments (F = 18.49; *p* < 0.001). Post-hoc Tukey’s tests were used to identify significant variations relative to the control conditions or among the different treatments (the latter are indicated by horizontal lines) (* *p* ≤ 0.05).

**Figure 9 cancers-13-05460-f009:**
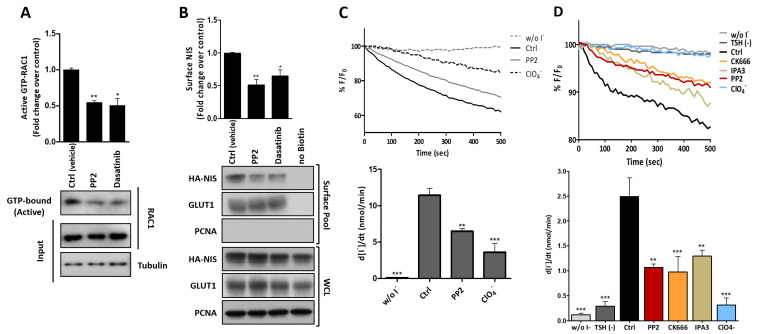
SRC functions upstream of RAC1 to promote NIS residency at the PM. (**A**) HA-NIS-TPC1 were treated with either the vehicle or one of two SRC inhibitors, PP2 (2 µM for 1 h) or Dasatinib (150 nM for 1 h), and the endogenous RAC1 activation status was assessed by monitoring the levels of active, GTP-bound RAC1 with CRIB-domain pull-down assays. The endogenous levels of the total (input) and GTP-bound RAC1 in the pulled-down fraction were assessed by WB using an anti-RAC1 antibody. The total lysates were further probed for tubulin as the loading control. Plotted is the densitometry analysis of the WB bands (means ± SEM of three independent assays), using ImageJ software. One-way ANOVA analysis detected significant differences between the treatments (F = 19.86; *p* = 0.0186). Post-hoc Dunnett’s tests were used to identify significant variations relative to the control (Ctrl) conditions (* *p* ≤ 0.05; ** *p* ≤ 0.01). (**B**) HA-NIS-TPC1 cells treated as in (**A**) were analyzed by surface protein biotinylation. The surface fraction or correspondent whole-cell lysates (WCL) were analyzed by WB, as indicated. The ‘no Biotin’ condition, corresponding to cells that were not incubated with biotin, was used to control the specificity of the biotinylated protein pull-down. PCNA expression served as the loading (WCL) and intracellular protein contamination control (Surface pool). GLUT-1 served as the positive loading control for cell surface protein extracts. The WB bands were quantified as in (**A**), and were plotted as the means ± SEM of at least three independent assays. A one-way ANOVA analysis detected significant differences between the treatments (F = 14.22; *p* = 0.0086). Post-hoc Dunnett’s tests were used to identify significant variations relative to the control (Ctrl) (* *p* ≤ 0.05; ** *p* ≤ 0.01). (**C**) Representative traces of the iodide-induced YFP fluorescence decay of the HA-NIS/HS-YFP-TPC1 cells treated with either the vehicle, PP2 (2 µM for 1 h), or ClO_4_^−^ (50 mM for 10 min), which were continuously recorded for 500 s after exposure to 50 mM iodide (upper graph). The iodide influx rates (lower graph) were calculated by fitting the curves to the exponential decay function. The data are means ± SEM of four independent assays. The significant variations between the treatments were assessed by one-way ANOVA (F = 43.61; *p* < 0.001) followed by Dunnett’s post- hoc test, as compared with the control conditions (Ctrl) (** *p* ≤ 0.01; *** *p* < 0.001). (**D**) PCCL3 cells stably expressing the HS-YFP iodide sensor were serum-starved for 24 h, followed by stimulation or not with TSH (1 mU/mL for 48 h). The cells were then treated for 1 h with either vehicle, PP2 (2 µM), IPA3 (10 µM), or CK666 (100 µM), or for 10 min with 1 mM of ClO_4_^−^, and the YFP fluorescence was recorded continuously, as described in the legend to Figure 1. The data are the means ± SEM of at least three independent assays. A one-way ANOVA analysis detected significant differences between the treatments (F = 10.61 and *p* < 0.001). Post-hoc Dunnett’s tests were used to identify significant variations relative to the control (Ctrl) conditions (** *p* ≤ 0.01; *** *p* ≤ 0.001).

**Figure 10 cancers-13-05460-f010:**
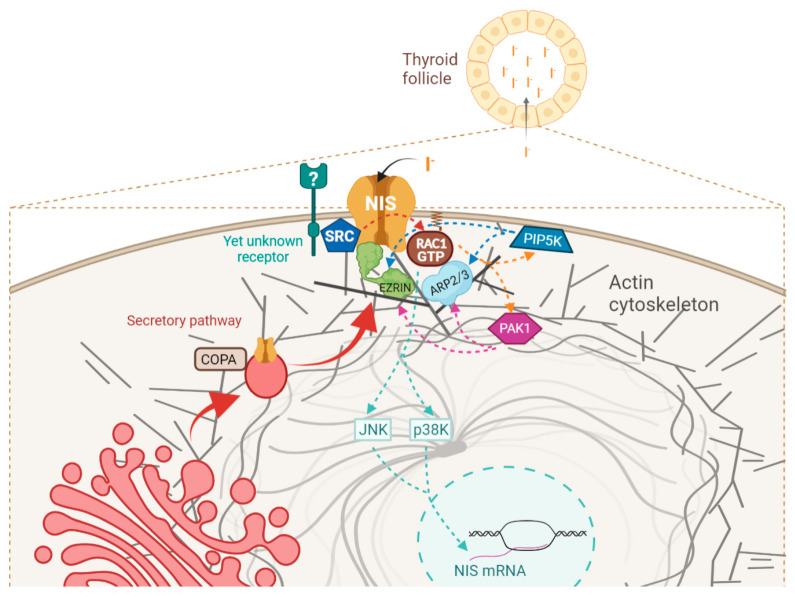
Diagram depicting the current model for the regulation of NIS functional residency at the PM by the SRC/RAC1/PIP5K/PAK/EZRIN/NIS pathway. In brief, NIS localization and function at the PM depends on its binding to SRC kinase, the activity of which, triggered by yet-unknown factors, leads to the recruitment and activation of the small GTPase RAC1. RAC1 signals not through p38 nor through JNK (two kinases downstream of RAC1 known to participate in the transcriptional regulation of NIS [59,60]), but through PAK1 and PIP5K. The combined activity of these kinases promotes ARP2/3-mediated actin polymerization, and the recruitment and binding of the actin anchoring protein EZRIN to NIS, promoting its residency and function at the PM of normal and TC cells. The figure was created with BioRender.com, accessed on 26 July 2021.

**Table 1 cancers-13-05460-t001:** Top five best-represented biological processes (BP) among the candidate NIS whole cell (WC) and plasma membrane (PM) interactors. Annotated GO-term enrichment analysis among the MS identified proteins with putative direct interactions with NIS. The analysis was performed using the STRING algorithm (https://string-db.org/, accessed on 3 May 2021) on the subsets of proteins exclusively detected in either the WC and PM datasets, and those present in both datasets.

Exclusively at NIS-PM Dataset				Common to NIS-PM and NIS-WC Datasets				Exclusively at NIS-WC Dataset			
Number of Proteins	109			Number of Proteins	19			Number of Proteins	33		
GO Term	Description	Count by Term	FDR	GO Term	Description	Count by Term	FDR	GO Term	Description	Count by Term	FDR
GO:0007010	cytoskeleton organization	29	4.51 × 10^−11^	GO:0050896	response to stimulus	17	4.00 × 10^−3^	GO:0006810	transport	21	7.26 × 10^−6^
GO:0030029	actin filament-based process	24	1.67 × 10^−12^	GO:0006810	transport	14	1.90 × 10^−3^	GO:0046903	secretion	12	6.47 × 10^−6^
GO:0030036	actin cytoskeleton organization	21	3.77 × 10^−11^	GO:0009888	tissue development	9	3.80 × 10^−3^	GO:0002252	immune effector process	11	1.18 × 10^−5^
GO:0097435	supramolecular fiber organization	17	3.39 × 10^−8^	GO:0071453	cellular response to oxygen levels	4	4.20 × 10^−3^	GO:0001775	cell activation	11	2.84 × 10^−5^
GO:0007015	actin filament organization	16	4.51 × 10^−11^	GO:0031952	regulation of protein autophosphorylation	3	4.00 × 10^−3^	GO:0043312	neutrophil degranulation	10	5.22 × 10^−6^

## Data Availability

The data is contained within the article or Appendix A.

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
