# Peer review of "Analysis of NIS Plasma Membrane Interactors Discloses Key Regulation by a SRC/RAC1/PAK1/PIP5K/EZRIN Pathway with Potential Implications for Radioiodine Re-Sensitization Therapy in Thyroid Cancer"

_cancers, 2021, doi:10.3390/cancers13215460_

Round 1
Reviewer 1 Report
The authors in this manuscript report on “Analysis of NIS plasma membrane interactors discloses key regulation by a SRC/RAC1/PAK1/PIP5K/EZRIN pathway with potential implications for radioiodine re-sensitization therapy in thyroid cancer”. This manuscript is focused on understanding the clinical and functional impact of NIS in RAI-refractory DTC. Studies indicating that NIS mRNA increased in refractory DTC cells, but some only reduced levels of iodide uptake. Using systematic biology as mass spectrometry and gene ontology analysis, this study identify a pathway by which the PM localization and function of NIS depends on its binding to SRC kinase, which leads to the recruitment and activation of the small GTPase RAC1. RAC1 signals through PAK1 and PIP5K to promote ARP2/3-mediated actin polymerization, and the recruitment and binding of the actin anchoring protein EZRIN to NIS, promoting its residency and function at the PM of normal and TC cells. The results are indicating regulation of NIS localization and function at the PM of TC cells. In conclusion, new venues for therapeutic intervention in TC, modulating abnormal SRC signaling in refractory TC, from a proliferative/invasive effect to the re-sensitization of TC to RAI by inducing NIS retention at the PM was introduced.
Thank you for the efforts in revisions. It seems that the rebuttal was not well organized, which makes it difficult to follow your statement. Additionally, since you conducted some additional experiments in the revised manuscript, many of my major concerns were not successfully addressed.
- Validation of high-confidence NIS interactions detected by MS analysis showed in Figure 3. Do these blots run separately and then merged between the IP and Input. Data is differential band pattern
- as well as in Figure S6.
- Effects of IQGAP, COPA, RAC1 and EZRIN depletion on NIS abundance at the PM showed in Figure 4. Do these blots run separately and then merged between the control, siCOPA, and non-biotin. Data is differential band pattern as well as in Figure S4.
- Better resolution and quality of Figure 2A/C and 4D/F are desirable.
Author Response
The authors would like to thank the reviewers for their second revision of our manuscript. We also appreciate all the constructive comments and suggestions, which helped us further improve the clarity and quality of the manuscript.
Below, we provide a point-by-point response to the reviewer's comments, indicating the changes made to the manuscript text to address them. All changes to the manuscript text were marked up using the “Track Changes” function of MS Word, as again requested by the editors.
Reviewer 1
The authors in this manuscript report on “Analysis of NIS plasma membrane interactors discloses key regulation by a SRC/RAC1/PAK1/PIP5K/EZRIN pathway with potential implications for radioiodine re-sensitization therapy in thyroid cancer”. This manuscript is focused on understanding the clinical and functional impact of NIS in RAI-refractory DTC. Studies indicating that NIS mRNA increased in refractory DTC cells, but some only reduced levels of iodide uptake. Using systematic biology as mass spectrometry and gene ontology analysis, this study identify a pathway by which the PM localization and function of NIS depends on its binding to SRC kinase, which leads to the recruitment and activation of the small GTPase RAC1. RAC1 signals through PAK1 and PIP5K to promote ARP2/3-mediated actin polymerization, and the recruitment and binding of the actin anchoring protein EZRIN to NIS, promoting its residency and function at the PM of normal and TC cells. The results are indicating regulation of NIS localization and function at the PM of TC cells. In conclusion, new venues for therapeutic intervention in TC, modulating abnormal SRC signaling in refractory TC, from a proliferative/invasive effect to the re-sensitization of TC to RAI by inducing NIS retention at the PM was introduced.
Thank you for the efforts in revisions. It seems that the rebuttal was not well organized, which makes it difficult to follow your statement. Additionally, since you conducted some additional experiments in the revised manuscript, many of my major concerns were not successfully addressed.
- Validation of high-confidence NIS interactions detected by MS analysis showed in Figure 3. Do these blots run separately and then merged between the IP and Input. Data is differential band pattern.
The authors apologize but perhaps we did not fully understood the reviewer’s concerns in the previous revision. As we mentioned in the previous response, the images in Figure 3 are cutouts of the corresponding WBs shown in the “Original WB films” file, uploaded with the manuscript. There we show the complete WB membranes used for the figure, and it can be seen that the co-IPs and input extracts probed for the presence of the candidate proteins were run side-by-side on the same gel. The regions corresponding to the indicated protein bands were then cutout, namely to remove markings, and assembled in the figure.
Perhaps the reviewer’s concern relates to the fact that, although two input lanes are shown in the original WBs, only one input lane was shown in the figure. While both inputs had equivalent protein amounts, in hindsight, this may have been a poor layout option that generated some confusion. For this reason, we now extended the input cutouts in the figure to include both input lysates (used in Ctrl and α-HA IPs), and made the correspondent alterations to the labels in the “Original WB films” file.
2. as well as in Figure S6.
For consistency, we replaced the WB images in Figure S6A with those of an experimental replicate in which the two input lysates were run side-by-side with the co-IPs.
3. Effects of IQGAP, COPA, RAC1 and EZRIN depletion on NIS abundance at the PM showed in Figure 4. Do these blots run separately and then merged between the control, siCOPA, and non-biotin. Data is differential band pattern as well as in Figure S4.
As documented in the “Original WB films” file, the “no biotin” controls were run alongside with their corresponding siRNA conditions in the same gel, for both the cell surface extracts and the total lysates. These membranes were then initially probed with anti-HA and anti-PCNA (the latter used as contamination and loading control, respectively for surface and total extracts, as explained in the figure legends). These were also the membranes that we reprobed afterwards with anti-GLUT1, as the requested loading control for the cell surface extracts.
However, these gels also contained samples from other experimental conditions that were not relevant for the figure. Thus, when assembling the figure, the lanes containing the relevant samples (siCtrl and indicated siRNA) were spliced (as indicated by the line boxes) to their corresponding “no biotin” control from the same gel and WB film (see the “Original WB films” file). When, for technical reasons (such as overlapping band sizes or cross-reacting antibodies), the samples were run again to monitor siRNA efficiency (Fig. 4A), transgene overexpression (Fig S4A), or endogenous kinase phosphorylation (Fig. 4B), the relevant lysates (WCLs) were then loaded again side-by-side on a new gel, and thus the figure shows an unsliced image for those WCLs’ WBs.
- Better resolution and quality of Figure 2A/C and 4D/F are desirable.
The authors apologize but could see no decrease in image resolution in the indicated figures inserted into the original manuscript word file. Perhaps the resolution loss reported by the reviewer resulted from pdf compression? Notwithstanding, should the paper be accepted, we will make sure that all the figures are at adequate resolution before approving the manuscript proofs.
Reviewer 2 Report
In this revised version of the paper the authors addessed all the criticisms.
Author Response
The authors thank the reviewer for revising our manuscript a second time.
Reviewer 3 Report
The work has been improved. I appreciate authors’ efforts in providing new data and the requested controls.
-Reference 23 and 24 are duplicated in the revised version of the manuscript.
- The new PCCL3 data reported in Fig 9D lack of WB controls showing NIS protein expression and a proof of the three inhibitors efficacy. The reported data are in agreement with the authors' findings but are pretty far from being a strong biological validation. I thus recommend the authors to tone down their statement (line 687-689) “ These observations indicate the same pathway acts to promote the PM residency of endogenous NIS in PCCL3 cells, further validating the observations made in TPC1 cells”.
A more appropriate description is: “ These observations SUGGEST that the same pathway MAY ACT to promote the PM residency of endogenous NIS in PCCL3 cells, further SUPPORTING the observations made in TPC1 cells”.
-In addition, I would like to point out that for drug treatments (EHT1864, CK666, IPA3, PP2 or Dasatinib) formal evidence of pharmacological inhibition should be included in WBs, showing also the expression of the corresponding drug target and/or at least an affected downstream effector, and thus providing a reliable readout of the inhibitor efficacy. These are missing in Fig 4C-E, 6B-C, 7A and 9D. I recommend the authors to take into account this point for future studies.
Author Response
The authors would like to thank the reviewers for their second revision of our manuscript. We also appreciate all the constructive comments and suggestions, which helped us further improve the clarity and quality of the manuscript.
Below, we provide a point-by-point response to the reviewer's comments, indicating the changes made to the manuscript text to address them. All changes to the manuscript text were marked up using the “Track Changes” function of MS Word, as again requested by the editors.
Reviewer 3
The work has been improved. I appreciate authors’ efforts in providing new data and the requested controls.
- Reference 23 and 24 are duplicated in the revised version of the manuscript.
The duplicated references have been corrected as requested.
- The new PCCL3 data reported in Fig 9D lack of WB controls showing NIS protein expression and a proof of the three inhibitors efficacy. The reported data are in agreement with the authors' findings but are pretty far from being a strong biological validation. I thus recommend the authors to tone down their statement (line 687-689). “These observations indicate the same pathway acts to promote the PM residency of endogenous NIS in PCCL3 cells, further validating the observations made in TPC1 cells”.
A more appropriate description is: “These observations SUGGEST that the same pathway MAY ACT to promote the PM residency of endogenous NIS in PCCL3 cells, further SUPPORTING the observations made in TPC1 cells”.
The authors acknowledge the reviewer’s comment and have toned down the indicated statement as suggested.
- In addition, I would like to point out that for drug treatments (EHT1864, CK666, IPA3, PP2 or Dasatinib) formal evidence of pharmacological inhibition should be included in WBs, showing also the expression of the corresponding drug target and/or at least an affected downstream effector, and thus providing a reliable readout of the inhibitor efficacy. These are missing in Fig 4C-E, 6B-C, 7A and 9D. I recommend the authors to take into account this point for future studies.
Again, the authors acknowledge the reviewer’s comment. Indeed, whereas the recapitulation of the effect of these inhibitors by the siRNA experiments gave us confidence in our observations in TPC1 cells, in PCCL3 cells, despite the consistency of the results, we did not have sufficient time during the previous revision to validate the inhibitory effect of the drugs using siRNAs or, as the reviewer mentions, using downstream effectors as readouts. Unfortunately, we will also not be able to perform the entailed experiments in the 7 days given for this 2nd revision. However, in parallel experiments done by the team in the meantime, we had the opportunity to carry out CRIB pull-down assays in PCCL3 cells that show that 1h treatment with EHT1864 does inhibit endogenous RAC1 activation in these cells. Since RAC1 is the pivotal regulator in the proposed pathway, we believe this additional information further supports our observations in PCCL3 and have included the data as a supplemental panel in Fig. S3, referring to it in section 3.3. of the manuscript.
Notwithstanding, we do understand the pertinence of the reviewer’s recommendation and will work to implement the necessary assays for future studies, in PCCL3 and other thyroid cell lines.
Round 2
Reviewer 1 Report
The revised manuscript is ready to be published.
This manuscript is a resubmission of an earlier submission. The following is a list of the peer review reports and author responses from that submission.
Round 1
Reviewer 1 Report
In this study the authors exploit an in vitro model based on the thyroid cancer cell line TPC1 ectopically expressing the sodium-iodide symporter (NIS) tagged with HA to identify NIS interacting proteins both at cellular and plasma membrane level. The used experimental techniques include Co-IP, Mass spectrometry, Western blot, transfection and pharmacological treatments.
The study shows many data and experiments, but they are not always properly reported (some controls are missing) and further validation are required.
- The major concern is about HA-NIS TPC1 system: in any part of the work I can find a formal validation or references related to a previous validation for this vector. Even thought the authors report that triple HA tag localized in the extracellular loop, this itself does not exclude the possibility that the tag may affect NIS protein properties such as for instance protein folding, localization, function and/or interaction with other proteins. A formal validation is required, in which TPC1 cell where transfected with NIS full-length construct compared with HA-NIS construct and in both NIS localization and Iodide uptake capacity are at least assessed.
-In HA-NIS TPC1 the expression of NIS should be confirmed not only by HA but also by NIS specific primary antibody, that is available as described by the authors line 205 “rabbit anti-NIS (Protein-Tech)”.
- Western blot: a positive loading control relative to surface pool extracts should be included.
- Western blot: IP and the corresponding Input should be run and shown on the same gel, e.g. cropped panels Figure 3B are not properly reported. Original uncropped gel images should be provided.
- The authors propose a model in which NIS residency at plasma membrane is regulated by SRC/RAC1/PIP5K/PAK/EZR pathway, but the reported experiments show the expression of HA-NIS in relation with one or at most two of these players at time, never all together. The concurrent expression of these players should be added in WB and possibly also by co-IF to assess a possible co-localizazion at plasma membrane level
- The proposed interacting pathway requires biological validation, for instance by assessing the expression of the identified players in thyrocytes (w/wo TSH stimulation, known to induce NIS expression, plasma membrane localization and function in normal thyroid cells). Only a small test has been performed by the authors on PCCL3 cells treated with RAC1 inhibitor, that is not sufficient to provide biological validation for all the other proposed interacting players.
Minor:
-From line 201 primary Ab catalog number or ID should be included in the method section
Reviewer 2 Report
The article "Analysis of NIS plasma membrane interactors discloses key regulation by a SRC/RAC1/PAK1/PIP5K pathway with potential implications for radioiodine re-sensitization therapy in thyroid cancer" is a valuable work conducted by Faria et al. opening new venues for therapeutic intervention in thyroid cancer, by modulating abnormal SRC 43 signaling in refractory thyroid cancer, from a proliferative/invasive effect to the re-sensitization of these tumors 44 to RAI therapy by inducing NIS retention at the PM.
Originality and a well conducted research will sustain the acceptance for publication in current form.
Reviewer 3 Report
In this paper the authors investigated the molecules and pathways implicated in NIS residency at the plasma membrane. They showed that NIS localization and function at the plasma membrane depend on the activation of SRC/RAC1/PAK1/PIP5K pathway, able to promote actin polymerization and the binding of the actin anchoring protein EZRIN to NIS. These findings may provide new therapeutic options in radioiodine refractory TC.
The paper is well written and experiments were elegantly performed, nevertheless I have some comments:
Did the authors validate the interaction of NIS with ACTB, SRC and ARPC4 at plasma membrane by Western blot?
In the legend of figure 4, the authors stated that the pathways identified promote NIS residency and function at the plasma membrane of normal and thyroid cancer cells. However, experiments were performed on TPC1 cell line and further experiments in other thyroid cancer and normal cell lines will be necessary.
Considering the positive associations between I131dose, SRC expression and patient survival (specify time) detected from TGCA dataset, the authors assume that a higher SRC levels could favor iodide uptake. However, the higher I131 dose required in cases with higher SRC expression may rather indicate a worst iodine uptake. Does SRC expression correlate with patient survival time? Moreover, it is surprising that the I131 dose does not correlate with NIS expression.
Minor comments:
Uniform the name of EZRIN all along the text.
In the Legend of figure 7, change 7D with 7C.
In supplemental table 5 provide measuring units for expression and survival time. In the legend specify that correlations were indicated between I131 dose and gene expression, too.
Reviewer 4 Report
Dear Cancers,
Thank you for the opportunity to review this manuscript. This article focused on the mechanism of post-translational failure in the delivery of NIS to the plasma membrane or an impaired residency resulting in RAI-refractory differentiated thyroid cancers. They conducted an intact-cell labeling/immunoprecipitation methodology to selectively purify NIS-containing macromolecular complexes from the plasma membrane. They proposed the NIS PM complexes were particularly enriched in proteins associated with the regulation of the actin cytoskeleton through applying gene ontology analysis to the mass spectrometry results, comparisons of the composition of NIS PM complexes to whole cell lysates. And they also identify novel insights into the regulation of NIS localization and function at the PM of DTC cells by a SRC/RAC1/PAK1/PIP5K pathway. Further therapeutic intervention in DTC from a proliferative/invasive effect to the re-sensitization of these tumors to RAI therapy by inducing NIS retention at the PM.
I have some questions for the authors about this article.
-
I have some questions for the authors about this article.
- The rationale of using PCNA as loading control since it's regarding to the nuclear expression and the absence in either WC- or PM-NIS co-precipitates, how about the whole cell expression?
- I would like to view all the validations of STRING -generated subnetwork with the 29 top-scoring NIS-PM candidates shown in Fig. 3A.
- Are the effects of iodide uptake inhibition from CIO4 dose dependent (shown in Fig. 1E & 5C & 8B)?
- The stars indicated significance in Fig. 6A/B/D & Fig. 7B were not so clear.
- The legend of Fig. 7C and the label of Fig. 7D were missing.
- Have you tried systemic biology for the unknown receptor mentioned in Fig. 10?
